# GEPS: Boosting Generalization in Parametric PDE Neural Solvers through Adaptive Conditioning

**Armand Kassaï Koupaï** [1] *    **Jorge Mifsut Benet** [1]    **Yuan Yin** [2]
**Jean-Noël Vittaut**[3]    **Patrick Gallinari**[1, 4]

[1] Sorbonne Université, CNRS, ISIR, 75005 Paris, France    [2] Valeo.ai, Paris, France
[3] Sorbonne Université, CNRS, LIP6, 75005 Paris, France    [4] Criteo AI Lab, Paris, France

## Abstract

Solving parametric partial differential equations (PDEs) presents significant challenges for data-driven methods due to the sensitivity of spatio-temporal dynamics to variations in PDE parameters. Machine learning approaches often struggle to capture this variability. To address this, data-driven approaches learn parametric PDEs by sampling a very large variety of trajectories with varying PDE parameters. We first show that incorporating conditioning mechanisms for learning parametric PDEs is essential and that among them, *adaptive conditioning*, allows stronger generalization. As existing adaptive conditioning methods do not scale well with respect to the number of parameters to adapt in the neural solver, we propose GEPS, a simple adaptation mechanism to boost GEneralization in Pde Solvers via a first-order optimization and low-rank rapid adaptation of a small set of context parameters. We demonstrate the versatility of our approach for both fully data-driven and for physics-aware neural solvers. Validation performed on a whole range of spatio-temporal forecasting problems demonstrates excellent performance for generalizing to unseen conditions including initial conditions, PDE coefficients, forcing terms and solution domain. *Project page*: https://geps-project.github.io/

## 1 Introduction

Solving parametric partial differential equations, i.e. PDE in which certain parameters—such as initial and boundary conditions, coefficients, or forcing terms—can vary, is a crucial task in scientific and engineering disciplines, as it plays a central role in enhancing our ability to model and control systems, quantify uncertainties, and predict future events (Cohen & Devore, 2015). Neural networks (NNs) are increasingly employed as surrogate models for solving PDEs by approximating their solutions (Long et al., 2018; Lu et al., 2021; Li et al., 2021). A primary challenge with these data-driven solvers is their ability to generalize across varying contexts of the physical phenomena. This is especially pronounced for dynamical systems, which can exhibit significantly different behaviors when subject to small changes in PDE parameters.

The usual approach for solving a parametric PDE with neural networks involves sampling instances from the PDE parameter distribution (e.g., the PDE coefficients), and then sampling trajectories for each instance of the underlying PDE (Takamoto et al., 2022). The training set thus consists of multiple trajectories per parameter instance, with the objective of generalizing to new instances. This approach aligns with the classical empirical risk minimization (ERM) framework: it assumes that the training dataset is large enough to represent the distribution of the dynamical system behaviors and that this distribution is i.i.d. (Bodnar et al., 2024). However, given the complexity of dynamical

---

*Corresponding author: armand.kassai@isir.upmc.fr

38th Conference on Neural Information Processing Systems (NeurIPS 2024).

systems and the variety of PDE terms and parameters, these assumptions are rarely met in practice. To address this difficulty, some methods relax these assumptions. For instance, certain approaches embed the PDE parameters as inputs, aiming to generalize to new dynamics (Brandstetter et al., 2023; Takamoto et al., 2023). While this can alleviate the issue, it does not facilitate generalization beyond the training distribution. Other approaches focus on few-shot settings, where a pre-trained model is fine-tuned for each new context (Subramanian et al., 2023; Herde et al., 2024). As we will see, fine-tuning NN solvers typically requires a substantial amount of data, when it is often scarce in practice. In general, given the complexity of physical phenomena, it is often unrealistic to expect a sample distribution that is sufficiently representative to enable reliable generalization to new instances of the same phenomenon.

Our claim is that neural PDE solvers should be explicitly trained with an *adaptive conditioning* mechanism to enable generalization to new parameter instances. For practical applications, this mechanism should condition the network on new dynamics using only a small amount of data. We assume access to several environments governed by the same general PDE, each defined by a unique set of parameters, from which trajectories can be sampled. This assumption aligns with approaches in multi-task learning (Yin et al., 2022) and meta-learning (Wang et al., 2022; Kirchmeyer et al., 2022), where network weights are conditioned on a context that encapsulates environment-specific information. Although various adaptive conditioning mechanisms have been proposed (Finn et al., 2017; Kirchmeyer et al., 2022; Wang et al., 2022), they are often limited to specific neural network architectures (Karimi Mahabadi et al., 2021) and struggle to scale effectively with data size or with the number of environments. We then propose an adaptation method that can handle a diversity of physical dynamics, is compatible with various NN backbones, and scales effectively with data size and the number of environments.

After introducing the problem in Section 2, we emphasize in Section 3 the need for adaptive conditioning to solve parametric PDEs and illustrate the limitations of traditional ERM based approaches. We then introduce GEPS, a low-rank $1^{\text{st}}$-order gradient-efficient adaptation mechanism for learning parametric PDE solvers. At inference, GEPS adapts to a new unseen environment, by learning a compact context vector $c^e$. This approach is instantiated in two representative settings: (i) purely agnostic approaches, where the solver is trained without any prior physical knowledge, and (ii) physics-aware approaches that combine differentiable numerical solvers with NN components in a hybrid framework. Finally section 5 compares GEPS with alternative adaptation baselines. Our contributions are as follows:

- We demonstrate on example PDEs that adaptive conditioning generalizes to multiple and new contexts, while classical ERM approaches fail to handle parametric PDEs.
- We propose an effective and scalable adaptive conditioning framework for learning neural solvers. It is based on a first-order meta-learning approach and leverages low-rank adaptation, enabling adaptation to unseen environments in a few shot context. This formulation is versatile enough to encompass both pure data-driven and physics-aware models.
- We provide experimental evidence of the performance and versatility of the approach on representative families of parametric PDEs, incorporating changes in initial conditions, PDE coefficients, forcing terms, and domain definitions, thereby covering both in-distribution and out-of-distribution scenarios.

## 2 Problem Description

We consider parametric time-dependent PDEs and aim to train models that generalize across a wide range of the PDE parameters, including initial conditions, boundary conditions, coefficient parameters, and forcing terms. For a given PDE, an environment $e$ is an instance of the PDE characterized by specific parameter values. We assume that all environments share common global features, such as the general form of the dynamics, while each environment $e$ exhibits some unique and distinct behaviors. A solution $\boldsymbol{u}^e(x, t)$ of the PDE in environment $e$ satisfies the PDE:

$$\frac{\partial \boldsymbol{u}^e(x, t)}{\partial t} = F^e\left(\boldsymbol{u}^e, \frac{\partial \boldsymbol{u}^e}{\partial x}, \frac{\partial^2 \boldsymbol{u}^e}{\partial x^2}, \dots, \boldsymbol{\mu}, \boldsymbol{f}\right), \quad \forall x \in \Omega, \forall t \in (0, T] \tag{1}$$

$$\mathcal{B}(\boldsymbol{u}^e)(x, t) = 0, \quad \forall x \in \partial\Omega, \forall t \in (0, T] \tag{2}$$

$$\boldsymbol{u}^e(x, 0) = \boldsymbol{u}_0^e, \quad \forall x \in \Omega \tag{3}$$

where $F^e$ is a function of the solution $\boldsymbol{u}^e$, its spatial derivatives, the PDE coefficients $\boldsymbol{\mu}$ and forcing terms $\boldsymbol{f}(x,t)$; $\mathcal{B}$ is the boundary condition (e.g., spatial periodicity, Dirichlet, or Neumann) that must be satisfied at the domain boundary $\partial\Omega$ and $\boldsymbol{u}_0$ is the initial condition (IC) sampled with a probability measure $\boldsymbol{u}_0 \sim \boldsymbol{p}^0(.)$. Environment $e$ is thus defined by a set of parameters $\boldsymbol{\xi} = \{\mathcal{B}, \boldsymbol{p}^0, \boldsymbol{\mu}, \boldsymbol{f}\}$.

The targeted task is dynamics forecasting, where a neural solver is auto-regressively rolled out on a temporal horizon $t \in [0, T]$. The goal is to approximate the evolution operator $F^e$ with a neural solver $\mathcal{G}_\theta(\boldsymbol{u}^e(x,t))$ parametrized by $\theta \in \mathbb{R}^{d_\theta}$, capable of generalizing to various PDE instances $\boldsymbol{\xi}$, both within (in-distribution) and outside (out-of-distribution) the training parameter distribution.

Therefore, we posit the observation of a set of environments $e$, each characterized by its specific PDE parameters $\boldsymbol{\xi}$; trajectories are sampled for each environment, each characterized by an IC $\boldsymbol{u}_0^e$. We define $\mathcal{E}_{\mathrm{tr}}$ as the set of environments used to train our model. In each training environment, $N_{\mathrm{tr}}$ trajectories are available $\mathcal{D}_{\mathrm{tr}}^e = \{\boldsymbol{u}_i(x,t)\}_{i=1}^{N_{\mathrm{tr}}}$. For the *in-distribution* evaluation, the model has already learned conditioning context parameters (described later in section 4.1) for each training environment and is simply tested on new trajectories from the same environments using the appropriate context. For *out-of-distribution*, the model is evaluated on trajectories from new environments from an evaluation set $\mathcal{E}_{\mathrm{ev}}$ and is adapted. We then assume access to $N_{\mathrm{ad}}$ trajectories $\mathcal{D}_{\mathrm{ev}}^e = \{\boldsymbol{u}_i(x,t)\}_{i=1}^{N_{\mathrm{ad}}}$ from the new environments to adapt the network. In our experiments we consider a scarce data, few-shot scenario where $N_{\mathrm{ad}} = 1$. After adaptation, we test our model on new unseen trajectories from these evaluation environments $\mathcal{E}_{\mathrm{ev}}$. This setting is illustrated in Figure 1.

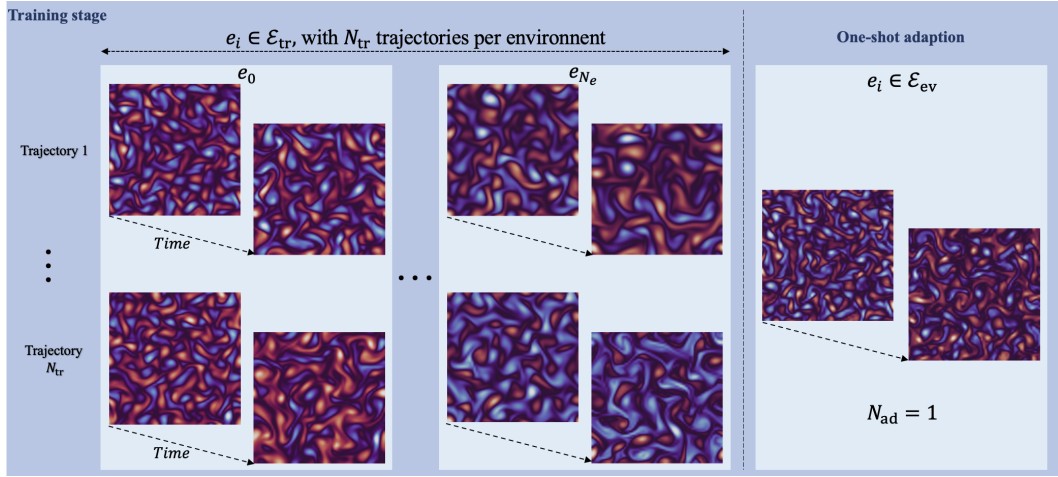

Figure 1: Multi-environment setup for the Kolmogorov PDE. The model is trained on multiple environments with several trajectories per environment (left). At inference, for a new unseen environment it is adapted on one trajectory (right).

Before introducing our method, we first aim to motivate the need for adaptive conditioning in learning to solve parametric PDEs, while illustrating the limitations of the classical ERM setting.

## 3   Motivations: ERM versus adaptive approaches for parametric PDEs

The classical ERM approach learns a single model on the data distribution, assuming that the training dataset is large enough to approximate the true data distribution. In practice, data acquisition is often costly and even simple physical systems can demonstrate a large variety of behaviors due to changes in the parameters. This may lead to poor generalization especially with scarce data.

To illustrate this, we will compare the performance and behavior of classical ERM approaches and of our adaptive conditioning method for solving parametric PDEs, on two example datasets: the 2D Gray-Scott and the 1D Burgers equations. In these experiments, we generate for each PDE a series of environments by sampling only the physical coefficients of the PDE $\boldsymbol{\xi} = \{\boldsymbol{\mu}\}$ (more details in Appendix B). We consider in-distribution generalization and out-of-distribution generalization (respectively in sections 3.1 and 3.2, both for an initial value problem (IVP), a common setting where the initial condition $\boldsymbol{u}_0$ corresponds to the system state at one time $t_0$ only. We then consider an alternative setting, where the neural solver is conditioned over a sequence of past states instead of

one state only (section 3.3), this is denoted as "temporal conditioning". For all experiments, reported results correspond to the averaged relative L2 loss on 32 unseen trajectories per environment.

### 3.1 In-distribution generalization for IVP: classical vs. adaptive conditioning approaches

Let us first compare adaptive conditioning and ERM approaches, for in-distribution evaluation, when scaling the number of training environments and trajectories. The models are trained on a range of environments - corresponding to different coefficients of the underlying PDE - and evaluated on the same environments with different initial conditions. Here GEPS is implemented with a classical CNN, while the tests for ERM are performed with four different backbones: the same CNN as used for GEPS but without adaptive conditioning, FNO (Li et al., 2021), MP-PDE (Brandstetter et al., 2023) and Transolver (Wu et al., 2024). Additionally, we also compared with a reference foundation model "Poseidon" (Herde et al., 2024). This model has been pre-trained on a variety of IVP PDE equations and is fine-tuned on our data. Poseidon being trained on 2D data is thus evaluated only for the Gray-Scott equation. For all the baselines, we consider the classical IVP setting where only one initial state is given as input.

**Scaling w.r.t. the number of environments.** We first examine how the number of training environments affects generalization to unseen trajectories within the same range of environments. The models are trained on 4 and 1024 environments with 4 trajectories per environments corresponding to different initial conditions. The evaluation is performed on 32 new trajectories from the same set of environments. As shown in Figure 2, Transolver, FNO, MP-PDE, and CNN fail to capture the diversity of behaviors and their performance stagnate when increasing the number of training environments. Non-conditioned methods are not able to capture the diversity of behaviors for several environments, regardless of the backbone used, when using only an initial state as input. Poseidon on its side behaves much better on Gray-Scott and is able to capture this diversity of dynamics. Our adaptive conditioning approach (GEPS on the figures) performs significantly better than all the baselines, outperforming also the large Poseidon foundation model. We can also observe that GEPS benefits from being trained on a large amounts of environments, as its generalization performance improves with the number of training environments.

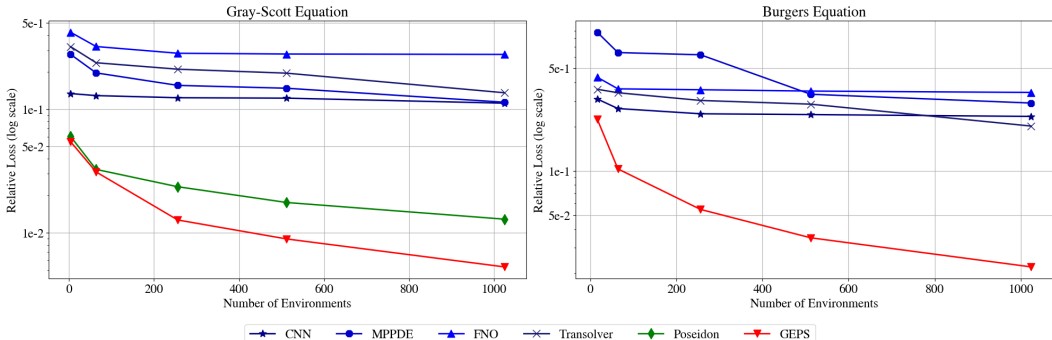

Figure 2: Comparison of ERM approaches (shades of blue) and Poseidon foundation model (green) with our framework GEPS (red) when increasing the number of training environments.

**Scaling w.r.t. the number of trajectories per training environment.** For this second series of experiments, we fix the number of environments at 4 and vary the number of training trajectories per environment from 4 up to 1024. Figure 3 shows the same behavior as for the previous experiments: ERM approaches rapidly reach a plateau and do not capture the variety of behaviors, even with a large number of trajectories, while GEPS scales well and improves with the number of training samples per environment. We additionally make a comparison with a CNN model trained and evaluated separately for each environment. We plot the average of the models' scores (indicated as "average" on the figure). This is an upper-bound of the performance that could be obtained with ERM models trained from scratch (no pretraining as for Poseidon). Note that this requires as many models as environments and is not scalable. While this performs significantly better than training over all the environments, GEPS matches or surpasses this approach.

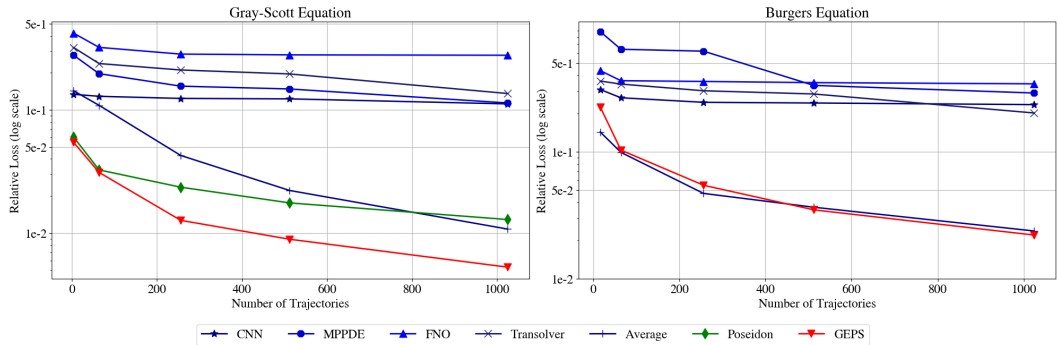

Figure 3: Comparison of ERM approaches (shades of blue) and Poseidon (green) with our framework GEPS (red) when increasing the number of trajectories per environment.

The experiments highlight that non-conditioned ERM approaches are unable to learn multi-environments datasets for solving IVP, whereas adaptive conditioning approaches like GEPS exhibit strong generalization performance, scaling with the number of training trajectories and environments.

## 3.2 Out-of-distribution generalization to new environments for IVP: classical vs. adaptive conditioning approaches

Let us now consider the out-of-distribution behavior of the two approaches. The models are trained on a sample of the environments from $\mathcal{E}_{tr}$ and their associated trajectories, in the same condition as for section 3.1. They are then evaluated on the trajectories of new environments. We report in figure 4 the out-of-distribution generalization performance of ERM methods and GEPS, for the Gray-Scott and Burgers equations, when pretrained on 4 (left) and 1024 (right) environments, with 4 trajectories per environment. For the test, one considers 4 new environments and evaluate on 32 trajectories per environment. Adaptation (GEPS) or fine tuning (baselines) is performed on one trajectory of a new environment. As for the baselines, we consider CNN and Transolver, plus the Poseidon foundation model for the 2-D Gray-Scott equation only. As above, one may observe a large performance gap between the non adaptive approaches and the adaptive GEPS. This supports our claim on the limitations of pure ERM based approaches to generalize to unseen dynamics configurations and new environments.

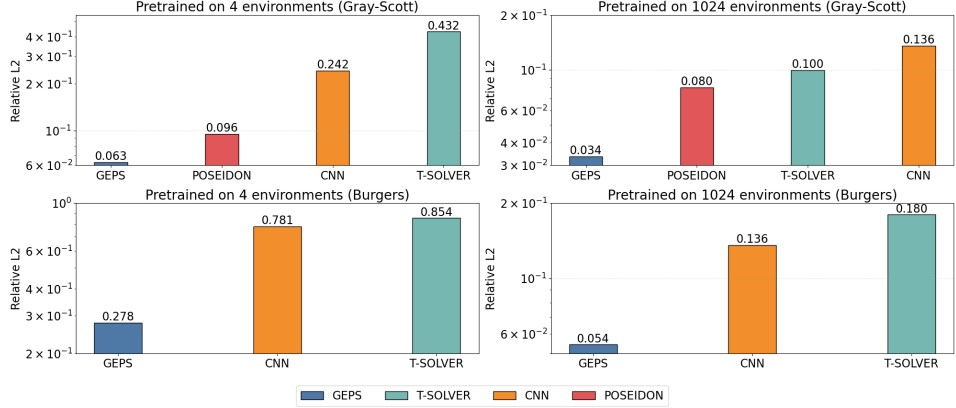

Figure 4: Out-distribution generalization on 4 new environments using one trajectory per environment for fine-tuning or adaptation. Models have either been pretrained on 4 environments (left column) or 1024 environments (right columns). Metric is Relative L2 loss.

## 3.3 In and out-of-distribution generalization performance with temporal conditioning

So far we have considered the classical IVP setting with only one initial state provided at time $t_0$. We consider now the situation where the model has access at $t_0$ to an history of past states and not to a

single initial state only. This is a common setting for learning PDE solvers: this allows the model to infer information on the dynamics and represents a more favorable case for the ERM baselines. We report in table 1 the in-distribution (*In-d*) and out-distribution (*Out-d*) distribution performance of ERM methods and GEPS, for Burgers and Gray-Scott PDEs. We did not consider the Poseidon (Herde et al., 2024) model that has been pre-trained only for one initial state IVPs. For in-distribution results, all methods are trained on 4 environments and evaluated on 32 new trajectories from the same environments. For out-of-distribution, adaptation (GEPS) and fine-tuning (baselines) is done on 1 trajectory per environment; 4 new environments are sampled and evaluation is performed on 32 new trajectories. We consider three different history sizes: 3, 5 and 10. Considering past history helps improve all the models. GEPS and the baselines show close performance for in-distribution, while GEPS is an order of magnitude better than the baselines for out-of-distribution.

Table 1: In-distribution and out-distribution results comparing different history window sizes. Metric is the Relative L2 loss.

| History → | 3 | | 5 | | 10 | |
|---|---|---|---|---|---|---|
| Method | *In-d* | *Out-d* | *In-d* | *Out-d* | *In-d* | *Out-d* |
| *Burgers equation* | | | | | | |
| Transolver | 1.95e-1 | 3.22e-1 | 1.12e-1 | 3.03e-1 | 5.64e-2 | 2.49e-1 |
| FNO | 4.28e-1 | 7.68e-1 | 3.07e-1 | 6.43e-1 | 6.13e-2 | 2.55e-1 |
| CNN | 1.84e-1 | 4.44e-1 | 1.62e-1 | 3.34e-1 | 3.16e-2 | 6.32e-2 |
| GEPS | **1.25e-2** | **1.63e-2** | **8.61e-3** | **1.04e-3** | **6.14e-3** | **8.73e-3** |
| *Gray-Scott equation* | | | | | | |
| Transolver | 1.82e-1 | 4.33e-1 | 9.88e-2 | 3.90e-1 | 9.57e-2 | 3.60e-1 |
| FNO | 1.86e-1 | 4.67e-1 | 1.76e-1 | 3.87e-1 | 1.93e-1 | 4.03e-1 |
| CNN | 7.12e-2 | 3.51e-1 | 5.96e-2 | 2.18e-1 | 6.54e-2 | 2.23e-1 |
| GEPS | **4.02e-2** | **5.78e-2** | **3.04e-2** | **5.02e-2** | **3.82e-2** | **5.14e-2** |

# 4 GEPS method

We introduce our framework for learning to adapt neural PDE solvers to unseen environments. It leverages a $1^{st}$ order adaptation rule and low-rank adaptation to a new PDE instance. We consider two settings commonly used for learning the PDE solvers. The first one leverages pure agnostic data-driven approaches, as already considered in section 3. The second one leverages incomplete physics priors and considers hybrid approaches that complement differentiable numerical solvers with deep learning components. The general framework is illustrated in Figure 5.

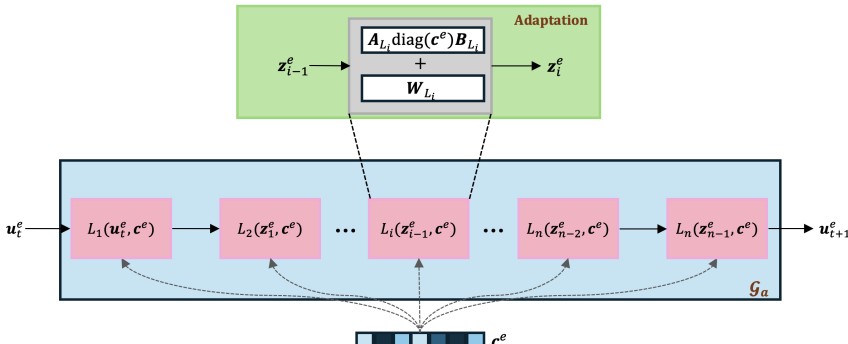

Figure 5: Our adaptation framework for our data-driven model. Block in blue refers to the data-driven module $\mathcal{G}_a$. Blocks $L_i$ in pink refer to the trainable modules. The green block describes the adaptation mechanism for the data-driven component, with $\boldsymbol{W}_{L_i}$ the weights of layer $L_i$. Context vector $\boldsymbol{c}^e$ conditions *all* the layers $\boldsymbol{W}_{L_i}$.

## 4.1 Adaptation rule

We aim to train a model $\mathcal{G}_\theta$ to forecast dynamical systems coming from multiple environments. We perform adaptation in the parameter-space: some parameters are shared across all the environments while others are environment-specific. Training consists in estimating the shared parameters and learning to condition the model on environment specific parameters. At test time, the shared

parameters are frozen and adaptation is performed on the environment specific parameters only. This setting is common to adaptation based approaches (Zintgraf et al., 2019; Kirchmeyer et al., 2022), however most of them do not scale to large problems while GEPS introduces an efficient scalable adaptation mechanism.

**Formulation**    We adapt the parameters of our model $\mathcal{G}_\theta$ using a low-rank formulation. Most deep-learning architectures can be decomposed into modules or layers - in our experiments we use MLPs, CNNs and FNOs. For simplification let us then consider a layer $L_i$ from $\mathcal{G}_\theta$ parameterized by a weight matrix $\boldsymbol{W}_{L_i} \in \mathbb{R}^{d_{in} \times d_{out}}$. Adaptation to an environment is performed through a low-rank matrix $\Delta \boldsymbol{W}_{L_i}^e = A_{L_i} \text{diag}(\boldsymbol{c}^e) B_{L_i}$, where $\boldsymbol{A}_{L_i} \in \mathbb{R}^{d_{in} \times r}, \boldsymbol{B}_{L_i} \in \mathbb{R}^{r \times d_{out}}, \boldsymbol{c}^e \in \mathbb{R}^r$. The weights of layer $L_i$ with the adaptation mechanism are then:

$$\boldsymbol{W}_{L_i}^e = \boldsymbol{W}_{L_i} + \boldsymbol{A}_{L_i} \text{diag}(\boldsymbol{c}^e) B_{L_i} \tag{4}$$

where $\text{diag}(\boldsymbol{c}^e)$ is a diagonal matrix capturing environment specific information and $\boldsymbol{A}_{L_i}, \boldsymbol{B}_{L_i}, \boldsymbol{W}_{L_i}$ are shared parameter matrices across all environments. Ideally, we want $\boldsymbol{c}^e$ to capture the number of degrees of variations for our environments. If for example our model $\mathcal{G}_\theta$ is an MLP, the adaptation mechanism for a linear layer $L_i$ corresponds to:

$$\boldsymbol{z}_i^e = (\boldsymbol{W}_{L_i} + \boldsymbol{A}_{L_i} \text{diag}(\boldsymbol{c}^e) \boldsymbol{B}_{L_i}) \boldsymbol{z}_{i-1}^e + \boldsymbol{b}_{L_i}^1 + \boldsymbol{b}_{L_i}^2 \boldsymbol{c}^e \tag{5}$$

where $\boldsymbol{W}_{L_i}, \boldsymbol{A}_{L_i}, \boldsymbol{B}_{L_i}, \boldsymbol{b}_{L_i}^1, \boldsymbol{b}_{L_i}^2$ are the parameters of layer $L_i$, shared across all environments. Only $\boldsymbol{c}^e$ is specific to each environment, but shared across all the layers of the network (cf Fig. 5).

Considering a low-rank adaption rule is popular in NLP, where large pre-trained models are adapted to new tasks by learning a low-rank matrix $\Delta \boldsymbol{W} = \boldsymbol{AB}$ added to the frozen weights $\boldsymbol{W}$ of the pre-trained model (Hu et al., 2022). Our approach differs in two ways from this setting: (i) the model is learned from scratch without pretraining, i.e., we learn parameters $\{\boldsymbol{W}, \boldsymbol{A}, \boldsymbol{B}, \boldsymbol{c}^e\}$ jointly, (ii) during adaptation, we can adapt to new environments $e \in \mathcal{E}_{ev}$ by optimizing only the context vector $\boldsymbol{c}^e$, where it is initialized as $\boldsymbol{c}^e = \bar{c}_{tr}$, with $\bar{c}_{tr}$ the averaged value of contexts learned during training. We experimentally show in Appendix D.3 that the classical Gaussian parameter initialization proposed in LoRA is inefficient in our context.

## 4.2    Two-step training procedure

This meta-learning framework operates in two steps: Training and Adaptation at inference. The goal is to learn an initial starting point during the training stage using a sample of environments, allowing adaptation to a new environment by adjusting a small subset of parameters based on a limited data sample from the new environment. Unlike many meta-learning gradient based approaches, GEPS does not involve an inner loop and is a $1^{st}$ order method. During the adaptation phase, all the parameters except the context parameters $\boldsymbol{c}^e$ are frozen and $\boldsymbol{c}^e$ is learned from the new environment sample. This approach ensures rapid adaptation by keeping $\boldsymbol{c}^e$ low-dimensional. For simplicity, we refer to parameters shared across environments as $\theta^s$ and parameters specific to each environment as $\delta\theta^e \triangleq \boldsymbol{c}^e$, and denote $\theta^e = \{\theta^s, \delta\theta^e\}$. The optimization problem can thus be formulated as follows:

$$\min_{\theta^s, \delta\theta^e} \sum_{\mathcal{D}_{tr}^e \in \mathcal{E}_{tr}} \mathcal{L}(\{\theta^s, \delta\theta^e\}, \mathcal{D}_{tr}^e)$$

$$\text{subject to } \delta\theta^e = \arg\min_{\delta\theta^e} \sum_{\mathcal{D}_{ev}^e \in \mathcal{E}_{ev}} \mathcal{L}(\{\theta^s, \delta\theta^e\}, \mathcal{D}_{ev}^e) \tag{6}$$

We separate training and adaptation steps into a training and an adaptation loop as described in the pseudo-code Algorithm 1.

## 4.3    Hybrid formulation for learning dynamics

We consider here an alternative problem to the above agnostic formulation. We assume that part of the physics is modeled through a PDE equation and shall be complemented with a statistical module. This is a common situation in many domains where prior physical knowledge is available, but only in an incomplete form. We follow the formulation in Yin et al. (2021) were starting from a complete PDE, we assume that part of the equation is known and will be modeled with a differentiable solver,

**Algorithm 1:** Training and adaptation for our method

*Training:*
**Input:** $\{\mathcal{D}_{\mathrm{tr}}^e\}_{e\in\mathcal{E}_{\mathrm{tr}}}$ with $\#\mathcal{D}_{\mathrm{tr}}^e = N_{\mathrm{tr}}$;
Choose proper initialization for $\theta^s$. Assign $\boldsymbol{c}^e \leftarrow 0$, $\theta^e = \{\theta^s, \boldsymbol{c}^e\}$
**while** *no convergence* **do**
   | Sample $\mathcal{D}_{\mathrm{tr}}^e$ from $\bigcup_{e\in\mathcal{E}_{\mathrm{tr}}} \mathcal{D}_{\mathrm{tr}}^e$
   | $\theta^e \leftarrow \theta^e - \eta\nabla_{\theta^e}\mathcal{L}(\{\theta^s, \boldsymbol{c}^e\}, \mathcal{D}_{\mathrm{tr}}^e)$
**end**
*Adaptation:*
**Input:** $\{\mathcal{D}_{\mathrm{ev}}^e\}_{e\in\mathcal{E}_{\mathrm{ev}}}$ with $\#\mathcal{D}_{\mathrm{ev}}^e = 1$;
Load pre-trained weights $\theta^s$. Assign $\boldsymbol{c}^e \leftarrow \bar{c}_{\mathrm{tr}}$
**while** *no convergence* **do**
   | Sample $\mathcal{D}_{\mathrm{ev}}^e$ from $\bigcup_{e\in\mathcal{E}_{\mathrm{ev}}} \mathcal{D}_{\mathrm{ev}}^e$
   | $\boldsymbol{c}^e \leftarrow \boldsymbol{c}^e - \eta\nabla_{\boldsymbol{c}^e}\mathcal{L}(\{\theta^s, \boldsymbol{c}^e\}, \mathcal{D}_{\mathrm{ev}}^e)$
**end**

while it is complemented with a deep learning component for modeling the unknown part. We also assume that the coefficients of the known part of the PDE are unknown and shall be estimated. We thus aim at solving both a direct problem (the NN parameters) and an inverse problem (the PDE coefficients of the known PDE part). We consider dynamics for a given environment of the form:

$$\frac{\partial \boldsymbol{u}^e(x,t)}{\partial t} = H(F^e(\boldsymbol{u}(x,t), \ldots), R^e(\boldsymbol{u}(x,t), \ldots)), \quad \forall x \in \Omega, \forall t \in (0, T] \tag{7}$$

$F^e$ and $R^e$ respectively represent the known and unknown physics of environment $e$ and $H$ is a function combining the two components which is unknown in practice. As for $\mathcal{G}_\theta$, our model of the evolution operator for this physics-aware setting, we will use a simple combination:

$$\mathcal{G}_\theta = \mathcal{G}_a \circ \mathcal{G}_p \tag{8}$$

where $\mathcal{G}_p$ encodes the physical knowledge and corresponds to the known part of the PDE physical model and $\mathcal{G}_a$ is the data-driven model term complementing $\mathcal{G}_p$. With this model, we use an auto-regressive formulation to generate the full trajectory, using a Neural ODE (Chen et al., 2018) as time-stepper for predicting the state $u_{t+\tau}$ as $u_{t+\tau} = u_0 + \int_{t_0}^{\tau} \mathcal{G}_\theta(u(\tau))\mathrm{d}\tau$, as illustrated in Figure 15. The model is trained directly from trajectories simulated with the full PDE (known + unknown PDE components) using the MSE loss $\mathcal{D}_{\mathrm{tr}}^e$ (more details in Appendix C):

$$\mathcal{L}(\theta, \mathcal{D}_{\mathrm{tr}}^e) = \sum_{j=1}^{N} \int_{t\in I, x\in\Omega} \|(\mathcal{G}_\theta(\boldsymbol{u}_j(x,t)) - H(F^e(x,t,\boldsymbol{u}_j(x,t)), R^e(x,t,\boldsymbol{u}_j(x,t)))\|_2^2 \mathrm{d}x\mathrm{d}t \tag{9}$$

The NN component is adapted as in section 4.1 through a $\boldsymbol{c}^e$ parameter context vector. For estimating the PDE coefficients, we considered two alternatives. One consists in using the learned code $\boldsymbol{c}^e$ for adapting the physical parameters $\theta_p^e = \theta_p + W_p\boldsymbol{c}^e$, where $\theta_p, W_p$ are shared parameters across all environments. The second one directly learns the parameters $\theta_p^e$ for each environment by gradient descent on the loss function. In the first approach, only context vectors $\boldsymbol{c}^e$ are learned while in the second approach, $\boldsymbol{c}^e$ and $\theta_p^e$ are learned jointly during adaptation. The performance of the two methods are similar. For the experiments, we evaluated both and used the better-performing one.

## 5 Experiments

### 5.1 Dynamical systems

We performed experiments on four dynamical systems, including one ODE and 3 PDEs. The ODE models the motion of a pendulum, which can be subject to a driving or damping term. We consider a Large Eddy Simulation (LES) version of the Burgers equation, a common equation used in CFD where discontinuities corresponding to shock waves appear (Basdevant et al., 1986). We additionally study two PDEs on a 2D spatial domain: Gray-Scott (Pearson, 1993), a reaction-diffusion system with complex spatio-temporal patterns and a LES version of the Kolmogorov flow, a 2D turbulence

equation for incompressible flows. For the pure data-driven approaches, we make no prior assumption on the underlying physics, while for the physics-aware hybrid modeling problem, we assume that the physical equation is partially known and that the deep learning component targets the modeling of the unknown terms (details on the known/unknown terms for each equation are provided in Appendix B). The setting is the classical IVP formulation when only one initial state $\boldsymbol{u}_0^e$ is provided.

## 5.2 Evaluation Setting

While in section 3 we compared GEPS with baselines ERM approaches and with a foundation model, our objective here is to assess the performance of GEPS w.r.t. alternative adaptation based approaches. We evaluate the model performance on two key aspects. • **In-distribution generalization**: the model capability to predict trajectories defined by unseen ICs on all training environments $e \in \mathcal{E}_{\text{tr}}$, referred as *In-d*. • **Out-of-distribution generalization**: the model ability to adapt to a new environment $e \in \mathcal{E}_{\text{ev}}$ by predicting trajectories defined by unseen ICs, referred as *Out-d*. Each environment $e \in \mathcal{E}$ is defined by changes in system parameters, forcing terms or domain definition. $d_p$ represents the degrees of variations used to define an environment for each PDE equation; $d_p = 4$ for the pendulum equation, $d_p = 3$ for the 1D Burgers, $d_p = 2$ and $d_p = 3$ for the Gray-Scott and the Kolmogorov flow equation respectively (more details in Table 4 in Appendix B). For each dataset, we collect $N_{\text{tr}}$ trajectories per training environment. For adaptation, we consider $N_{\text{ad}} = 1$ trajectory per new environment in $\mathcal{E}_{\text{ev}}$ to infer the context vector $c^e$. Evaluation is performed on 32 new test trajectories per environment. We report in Table 2 the relative MSE: $\frac{1}{N} \sum_{i=1}^{N} \frac{\|y_i - \hat{y}_i\|_2^2}{\|y_i\|_2^2}$.

## 5.3 Generalization results

**Implementation** We used a standard MLP for the Pendulum equation, a ConvNet for GS and Burgers equations and FNO for the vorticity equation. All activation functions are Swish functions. We use an Adam optimizer over all datasets. Contrary to Section 3, we perform time-integration with a NeuralODE (Chen, 2018) with a RK4 solver, as it was done for other multi-environments frameworks for physical systems (Yin et al., 2022; Kirchmeyer et al., 2022). Architectures and training details are provided in Appendix E.

**Baselines** As baselines, for the pure data-driven problem, we consider four families of multi-environment approaches. The first one consists in gradient-based meta-learning methods with CAVIA (Zintgraf et al., 2019) and FOCA (Park et al., 2023). The second one is a multi-task learning method for dynamical systems: LEADS (Yin et al., 2022). The third one is a hyper-network-based meta-learning method which is currently SOTA among the adaptation methods, CoDA (Kirchmeyer et al., 2022). As for the hybrid physics-aware problem, we implemented a meta-learning formulation of the hybrid method APHYNITY (Yin et al., 2021), where the adaptation is performed on the physical PDE coefficients only while the NN component is shared across all the environments. GEPS-Phy is our physics-aware model (section 4.3) were all the parameters, PDE coefficients and context vector $c^e$ are adapted at inference. We also implemented "Phys-Ad", where we use the same formulation as for the hybrid GEPS-Phy, but adaptation is performed on the coefficients of the physical component only while the neural network component is shared across all environments. All the baselines share the same network architectures than the ones used for GEPS, indicated in the implementation paragraph above. More details on baselines implementation are provided in Appendix E.6.

**In-distribution and out-of-distribution results** We report results for in and out-of-distribution generalization in table 2 for both the data-driven and the hybrid settings. Across all datasets, our framework performs competitively or better than the baselines for the two settings. Our method is able to correctly adapt to new environments in an efficient manner, updating only context parameters $c^e$. For the agnostic data-driven experiments, GEPS obtains the best results, although being on the same range of performance as other methods.

*The main differentiator of GEPS w.r.t. the baselines lies in its lower complexity.* In terms of training time, GEPS is way less expensive than gradient-based approaches like CAVIA that involves an outer and inner loop and LEADS, which learns a model specific to each environment. In terms of number of parameters, CoDA needs more training parameters because of its adaptation mechanism relying on a hyper-network. A comparison with the baselines in terms of parameter complexity is provided in table 10. Additional results in Appendix D, show that our data-driven framework adapts faster

Table 2: **In-distribution and Out-of-distribution results on 32 new test trajectories per environment**. For out-of-distribution generalization, models are fine-tuned on 1 trajectory per environment. Metric is the relative L2 loss. '−' indicates inference has diverged.

| Method | Pendulum ($\times 10^{-2}$) | | Gray-Scott ($\times 10^{-2}$) | | Burgers ($\times 10^{-3}$) | | Kolmogorov ($\times 10^{-1}$) | |
|---|---|---|---|---|---|---|---|---|
| | *In-d* | *Out-d* | *In-d* | *Out-d* | *In-d* | *Out-d* | *In-d* | *Out-d* |
| *Data-driven* | | | | | | | | |
| LEADS | 20.8 ± 1.01 | 51.1 ± 3.47 | 3.11 ± 0.25 | 3.81 ± 0.77 | 6.31 ± 0.52 | 64.1 ± 2.65 | 5.61 ± 0.37 | 9.18 ± 0.14 |
| CAVIA | 56.8 ± 9.73 | 91.8 ± 15.8 | 1.63 ± 3.82 | 23.1 ± 6.86 | 15.5 ± 1.06 | 225 ± 7.94 | 6.19 ± 0.02 | 8.48 ± 0.16 |
| FOCA | 41.0 ± 4.31 | 91.4 ± 9.70 | 1.71 ± 5.10 | 14.5 ± 3.34 | 92.2 ± 8.21 | 157 ± 23.6 | 6.30 ± 0.02 | 9.18 ± 0.14 |
| CoDA | 21.7 ± 1.08 | 66.2 ± 3.17 | 3.19 ± 0.07 | 2.89 ± 4.03 | 4.98 ± 0.19 | 74.5 ± 6.15 | 4.02 ± 0.57 | 6.34 ± 1.11 |
| *GEPS* | **20.8 ± 0.10** | **50.8 ± 3.90** | **2.22 ± 0.18** | **1.86 ± 2.11** | **2.59 ± 0.02** | **52.9 ± 5.87** | **2.94 ± 0.04** | **5.10 ± 0.11** |
| *Hybrid* | | | | | | | | |
| APHYNITY | 67.2 ± 9.68 | 69.0 ± 0.35 | **0.14 ± 0.04** | **0.20 ± 0.01** | 31.9 ± 0.08 | 307 ± 1.40 | − | − |
| Phys-Ad | 64.4 ± 1.10 | 58.4 ± 1.85 | 1.55 ± 1.41 | 1.82 ± 6.42 | 15.1 ± 0.10 | 89.9 ± 8.65 | − | − |
| *GEPS-Phy* | **8.04 ± 0.81** | **46.0 ± 3.64** | 0.67 ± 0.05 | 0.83 ± 0.05 | **5.43 ± 0.35** | **68.1 ± 3.75** | **2.78 ± 0.03** | **4.47 ± 0.17** |

than the strong baseline CoDA. Concerning the hybrid learning problem, incorporating physical knowledge leads to better results on all datasets except Burgers, where the fully data-driven method performs better. GEPS-Phy is better on all the datasets but Gray-Scott for which APHYNITY baseline is best. For Kolmogorov, the baselines did not converged at training.

## 5.4 Scaling to a larger dataset

To further illustrate the benefits of GEPS, we compare it to CoDA, the SOTA adaptation model, on a large dataset with a larger model than the ones used for the previous experiments. We use a dataset generated from a PDE with multiple differentiable terms that encompasses several generic PDEs, inspired from Brandstetter et al. (2022). The PDE writes as (more details in Appendix B):

$$[\partial_t u + \partial_x(\alpha u^2 - \beta \partial_x u + \delta \partial_{xx} u + \gamma \partial_{xxx} u)](t, x) = 0, \tag{10}$$

We report the results in table 3. All the methods are trained using a ResNet (He et al., 2016) architecture, using a context size $c^e$ of size 8. For *In-d*, we trained our model on 1200 environments with 16 training trajectories per environment and evaluated it on 16 new trajectories. For *Out-d*, we further adapt each model to 10 new environments given 1 context trajectory per environment and then evaluate it on 16 new trajectories per environment.

Table 3: **In-distribution and Out-distribution results**. Metric is the relative L2.

| Method | Combined ($\times 10^{-2}$) | | |
|---|---|---|---|
| | *In-d* | *Out-d* | *# Params* |
| GEPS | 7.52e-3 | 8.29e-2 | 4M |
| CoDA | 9.32e-2 | 1.78e-1 | 35M |

GEPS outperforms CoDA in terms of performance and number of parameters. This demonstrates the importance of scalable adaptation for few shot learning: GEPS is able to scale on large datasets using deep models while remaining parameter efficient.

## 6 Discussion and limitations

**Limitations** We have seen that adaptation is essential for learning to generalize neural PDE solvers, and that within this setting, integrating physical knowledge may help. Adaptation still requires sufficient training samples - both environments and trajectories. These findings are still to be confirmed for more complex dynamics and real world conditions for which the variety of behaviors should be much larger than for the simple dynamics we experimented with.

**Conclusion** We empirically demonstrated the importance of adaptation for generalizing to new environments and its superiority with respect to ERM strategies. We proposed a new $1^{st}$ order and low-rank scalable meta-learning framework for learning to generalize time-continuous neural PDE solvers in a few-shot setting. We also highlighted the benefits of directly embedding PDE solvers as hard constraints in data-driven models when faced with scarce environments.

# 7   Acknowledgments

We acknowledge the financial support provided by DL4CLIM (ANR-19-CHIA-0018-01), DEEPNUM (ANR-21-CE23-0017-02), PHLUSIM (ANR-23-CE23-0025-02), and PEPR Sharp (ANR-23-PEIA-0008, ANR, FRANCE 2030).

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

# A  Related Work

We review data-driven methods for learning parametric PDEs and existing works for incorporating physical priors in neural networks, in the context of dynamical systems.

## A.1  Learning parametric PDEs

**Traditional ML paradigm**  The majority of current approaches for learning parametric PDEs considers a traditional ERM approach by sampling from the PDE parameter distribution. In MP-PDE (Brandstetter et al., 2023), PDE parameters are directly embedded in the graph, allowing generalization to PDE parameters sampled from the same data distribution as those used for training. Neural operators (Kovachki et al., 2023; Takamoto et al., 2023) do similarly but require large training sets and often perform poorly for out-of-distribution (OOD) data. Alternatives to ERM such as invariant risk minimization (IRM) have shown stronger generalization by learning domain invariants (Arjovsky et al., 2020; Krueger et al., 2021). However, even for simple classification problems, IRM methods can fail to capture natural invariances and are extremely sensitive to sampling (Kamath et al., 2021).

**Multi-environment paradigm**  Several works have explored multi-environment/meta-learning settings for dynamical systems. • **Data-driven methods** like DyAd Wang et al. (2022) adapt dynamics models using a time-invariant context from observed state histories, which are often not accessible. Yin et al. (2022) developed LEADS, a multi-task method for dynamical systems that adapts in the functional space but requires training a new model for each environment. Park et al. (2023) proposed FOCA to address the limitations of second-order optimization of gradient-based meta-learning approaches (Zintgraf et al., 2019). (Kirchmeyer et al., 2022) make use of a hyper-network for fast adaptation to new dynamics by updating an environment dependent context encoding. While effective, gradient-based and hyper-network-based approaches respectively involve inner-loop updates or parameter increases with respect to context dimension. • **Model-based approaches** like physics-informed neural networks (PINNs) have been adapted for parametric PDEs (Cho et al., 2023); Huang et al. (2022) used meta-learning with PINNs through context vectors $c$ learned via auto-decoding. One drawback of PINN is their inability to handle scenarios where some physics are unknown.

## A.2  Hybrid learning

**Data-driven methods with soft constraints**  Physics losses as proposed by (Raissi et al., 2019) have been used as prior knowledge under the form of soft constraints for neural operators, with the objective to alleviate data scarcity issues (Wang et al., 2021; Li et al., 2023).

**Data-driven methods with hard constraints**  Alternatively, physical priors can be incorporated directly as hard constraints, leveraging the training of neural networks with differentiable physics. It has been particularly successful for accelerating computational fluid dynamics (CFD) by addressing numerical errors inherent in PDE discretization and for augmenting incomplete physics (Kochkov et al., 2021; Yin et al., 2021; Takeishi & Kalousis, 2021; Tathawadekar et al., 2023). Most of these approaches do not consider the parametric PDE setting.

# B  Dataset details

We present the equations and the data generation settings used for all dynamical systems considered in this work. In table 4, we report the different PDE parameters changed to generate training and adaptation environments.

## B.1  Damped and driven pendulum equation

We propose to study the damped and driven pendulum equation. The ODE represents the motion of a pendulum which can be subject to a damping and a forcing term. Even simple dynamical systems such as the pendulum equation can present a variety of complicated behaviors (e.g., under-damped, critically damped, resonance, super/sub-harmonic resonance, etc.), presented in figure 6.

Table 4: A description of the pivotal factors used to generate environments

| PDE | training ODE/PDE parameters | adaptation ODE/PDE parameters |
|---|---|---|
| Damped driven pendulum | $w_0 \in \{0.5, 0.7\}$ 
 $w_f \in \{0, 0.75, 1.0\}$ 
 $F \in \{0.0, 0.1, 0.2\}$ 
 $\alpha \in \{0., 0.2, 0.5\}$ | $w_0 \in \{0.5, 0.75, 1.0\}$ 
 $w_f \in \{0.3, 0.5, 0.7, 1.0\}$ 
 $F \in \{0.05, 0.1, 0.15, 0.2\}$ 
 $\alpha \in \{0.1, 0.5\}$ |
| 1D Burgers | $\nu \in \{5e{-}1, 5e{-}2, 5e{-}4\}$ 
 $f(x,t) = \{F(\sin(w_f x) + \cos(w_f t)), 0\}$ 
 $w_f = 1.5$ | $\nu \in \{1.0, 5e{-}5\}$ 
 $f(x,t) = F \exp(-w_f x^2)$ 
 $w_f = \{1.5, 3.0\}$ |
| 2D Gray-Scott | $F \in \{0.03, 0.039\}$ 
 $k \in \{0.058, 0.062\}$ | $F \in \{0.025, 0.042\}$ 
 $k \in \{0.050, 0.065\}$ |
| 2D Kolmogorov Flow | $\nu \in \{1e-3, 1e-4\}$ 
 $f(x,y,t) = \{F \sin(w_f y), 0\}$ 
 Domain $= \{[0, 0.75\pi], [0, \pi]\}$ | $\nu \in \{5e-4\}$ 
 $\{F \exp(w_f y^2), F(\cos(w_f x) + \sin(w_f t))\}$ 
 Domain $= \{[0, 1.25\pi]\}$ |

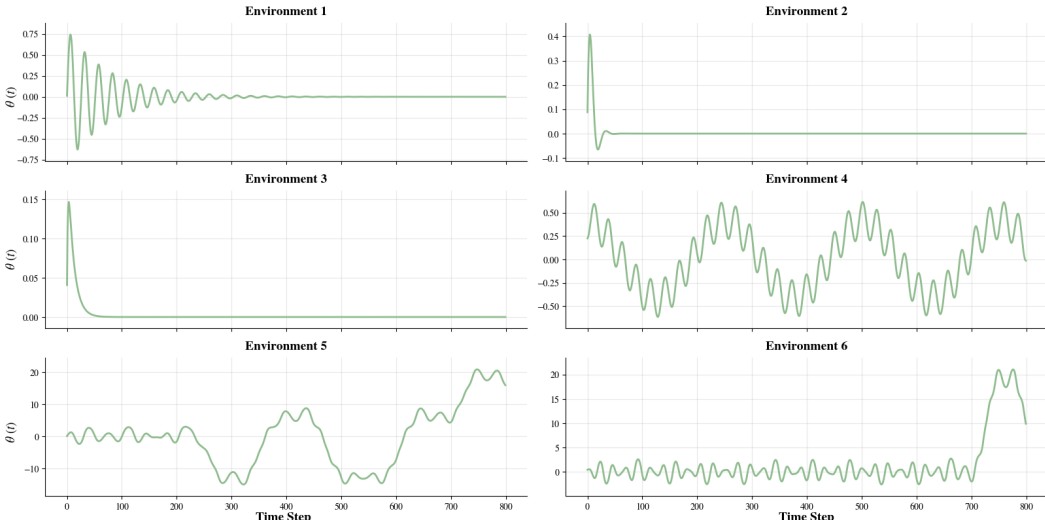

Figure 6: Visualization of different behaviors for the damped and driven pendulum equation

The form of the ODE is:

$$\frac{d^2\theta}{dt^2} + \omega_0^2 \sin\theta + \alpha \frac{d\theta}{dt} = f(t) \tag{11}$$

where $\theta(t)$ is the angle, $\omega_0$ the natural frequency, $\alpha$ the damping coefficient, $f(t)$ is a forcing function of the form $f(t) = F \cos(w_f t)$, where $w_f$ is the forcing frequency and $F$ the amplitude of the forcing. In our case where we suppose we only have incomplete knowledge of the phenomenon, we consider we do not know the damping term. For the initial condition, the pendulum is dropped from a distribution $\theta(t_0) \sim \mathcal{U}(0, \pi/12)$ and $\frac{d\theta}{dt}(t_0) \sim \mathcal{U}(0, 1)$.

**Data generation** Each environment is defined by changes in the system parameters resulting in different behaviors. In training environments, the pendulum is either subject to damping or forcing, e.g., environment 1 is a damped pendulum ($F = 0$) while environment 2 is a driven pendulum ($\alpha = 0$). We generate trajectories using a Runge-Kutta 8 solver. For training, we generated 4 distinct environments, each composed of 8 trajectories on the time horizon $[0, 25]$ with a time step $\Delta t = 0.5$. We also generate 32 trajectories on a longer time horizon $[0, 50]$ per training environment to evaluate in-distribution generalization. For adaptation, we evaluate our method on 4 distinct environments defined by parameters unseen during training. Only one trajectory is generated per environment with a time horizon is $[0, 25]$ to adapt the model to new dynamics; we evaluate the model's performance on 32 trajectories per environment on a time horizon $[0, 50]$. During adaptation, environments are defined such that trajectories are subject both to the forcing and damping term simultaneously.

## B.2 Gray-Scott equation

The PDE describes reaction-diffusion system with complex spatiotemporal pattern through the following 2D PDE:

$$\frac{du}{dt} = D_u \Delta u - uv^2 + F(1-u) \tag{12}$$

$$\frac{dv}{dt} = D_v \Delta v - uv^2 - (F+k)v \tag{13}$$

where $u, v$ represent the concentrations of two chemical components in the spatial domain $S$ with periodic boundary conditions. $D_u, D_v$ denote the diffusion coefficients respectively for $u, v$ and $F, k$ are the reactions parameters. We consider complete knowledge of the physical model, but instead uses a RK4 solver with $\Delta t = 1$ instead of using an adaptive RK45 solver. We generated environments by changing $F, k$ values, while $D_u, D_v$ are kept constant across all environments. For experiments in Section 3, we sampled uniformly values for $F$ and $k$ in ranges $[0.03, 0.04]$ and $[0.058, 0.062]$ respectively. For experiments in Section 5.3, details about the environments are given in table 4.

**Data generation** We generated trajectories on a temporal grid using an adaptive RK-45 solver. $S$ is a 2D space of dimension $32 \times 32$ with a spatial resolution of $\Delta s = 2$. For training, we generate 4 environments with one trajectory per environment defined on a temporal horizon $[0, 200]$. We evaluate in-distribution generalization on 32 trajectories per environment defined on a temporal horizon $[0, 400]$. For adaptation, we adapt to 4 new environments in one-shot learning manner. We evaluate the model's performance on 32 new trajectories per environments, defined on a temporal horizon $[0, 400]$.

## B.3 Burgers equation

Burgers' equation is a nonlinear equation which models fluid dynamics in 1D and features shock formation:

$$\frac{du}{dt} + \frac{du}{dx} - \nu \frac{d^2 u}{dx^2} + \mathcal{R}^{\text{closure}}(\bar{u}, u) = f(x, t) \tag{14}$$

where $u$ is the velocity field and $\nu$ is the diffusion coefficient and $f(x, t)$ is a forcing term. The unresolved scales $\mathcal{R}^{\text{closure}}(\bar{u}, u)$ and the forcing function is typically unknown, and needs to be directly learned from data. For the experiments done in Section 3, environments have been generated by sampling uniformly $\nu \in [1e-4, 0.5]$. For experiments in Section 5.3, we generated environments by changing the diffusion coefficient or changing the forcing function, as detailed in table 4.

**Data generation** For the DNS, we generate complex trajectories using a $5^{th}$ order central difference scheme using Runge-Kutta 45 solver with a time-step $\Delta t = 1e-5$ and $\Delta x = \frac{2\pi}{16384}$. Such trajectories are particularly costly to generate, therefore, we rather use LES. To obtain the ground truth LES trajectories, we apply a spatial filtering operator on the DNS trajectories. We also down-sample the temporal and spatial grid. Therefore, we obtain LES trajectories with a timestep $\Delta t = 1e-3$ and $\Delta x = \frac{1}{256}$.

For training, we generate 6 environments with 4 trajectories per environment. Trajectories have been generated on a temporal horizon $[0, 0.05]$ for training and for in-distribution evaluation. For adaptation, we adapt our model on 4 new unseen environments using only trajectory per environment. We evaluate out-distribution performance on 32 new ICs on a time horizon $[0, 0.05]$. For the generation of the data, we define a $[0, 2\pi]$ periodic domain and consider the following initial condition:

$$E(k) = \frac{2}{3}\sqrt{\pi} \left(\frac{k}{k_0}\right)^4 \frac{1}{k_0} \exp\left(-\left(\frac{k}{k_0}\right)^2\right) \tag{15}$$

Velocity is linked to energy by the following equation :

$$u(k) = \sqrt{2E(k)} \tag{16}$$

## B.4 Kolmogorov flow

We propose a 2D turbulence equation. We focus on analyzing the dynamics of the vorticity variable. The vorticity, denoted by $\omega$, is a vector field that characterizes the local rotation of fluid elements,

defined as $\omega = \nabla \times \mathbf{u}$. Like Burgers, we generate LES, leading to an unknown unclosed term appearing in the vorticity equation, expressed as:

$$\frac{\partial \omega}{\partial t} + (\mathbf{u} \cdot \nabla)\omega - \nu\nabla^2\omega + \mathcal{R}^{\text{closure}}(\bar{u}, u) = f(x, y, t) \tag{17}$$

Here, $\mathbf{u}$ represents the fluid velocity field, $\nu$ is the kinematic viscosity with $\nu = 1/Re$ and $f(x, y, t)$ is a forcing term. The unresolved scales $\mathcal{R}^{\text{closure}}(\bar{u}, u)$ and the forcing function is typically unknown, and needs to be directly learned from data. For the vorticity equation, environments can be defined either by changes in the viscosity term, the domain size or the forcing function term.

**Data generation** For the data generation of DNS, we use a 5 point stencil for the classical central difference scheme of the Laplacian operator. For the Jacobian, we use a second order accurate scheme proposed by Arakawa that preserves the energy, enstrophy and skew symmetry (Arakawa, 1966). Finally for solving the Poisson equation, we use a Fast Fourier Transform based solver. We discretize a periodic domain into $512 \times 512$ points for the DNS and uses a RK4 solver with $\Delta t = 5e - 3$. We obtain the ground truth LES by applying a spatial filtering operator on the DNS trajectories. We also down-sample temporal and spatial grid, thus obtaining LES trajectories with a timestep $\Delta t = 5e - 2$ on a $64 \times 64$ grid.

We generate 8 environments with 16 trajectories per environments for training. Trajectories have been generated on a temporal horizon $[0, 1]$ for training and $[0, 2]$ for in-distribution evaluation. During adaptation, we adapt our model on 4 unseen environments in a one-shot manner, using only one trajectory per environment. We evaluate out-distribution performance on 32 new ICs on a time horizon $[0, 2]$. We consider the following initial conditions:

$$E(k) = \frac{4}{3}\sqrt{\pi}\left(\frac{k}{k_0}\right)^4 \frac{1}{k_0}\exp\left(-\left(\frac{k}{k_0}\right)^2\right) \tag{18}$$

Vorticity is linked to energy by the following equation :

$$\omega(k) = \sqrt{\frac{E(k)}{\pi k}} \tag{19}$$

### B.5 Combined equation

We used the setting introduced in Brandstetter et al. (2023), but with the exception that we do not include a forcing term and add the $4^{th}$ order spatial derivative. The combined equation is thus described by the following PDE:

$$[\partial_t u + \partial_x(\alpha u^2 - \beta\partial_x u + \delta\partial_{xx}u + \gamma\partial_{xxx}u)](t, x) = 0, \tag{20}$$

$$u_0(x) = \sum_{j=1}^{J} A_j \sin(2\pi\ell_j x/L + \phi_j). \tag{21}$$

$\alpha, \beta, \delta$ and $\gamma$ are the parameters that are varied.

**Data generation** We used the spectral solver proposed in Brandstetter et al. (2022) to generate the solution. We sampled 1200 environments for training, by sampling uniformly within the ranges $\alpha \in [0.5, 1]$, $\beta \in [0, 0.5]$, $\delta \in [0, 1]$ and $\gamma \in [0, 1]$. For each parameter instance, we sampled 16 trajectories, resulting in 19200 trajectories. We then evaluate it on 19200 new trajectories. The trajectories were generated with a spatial resolution of 256 on a temporal horizon $[0, 30]$. We only keep 10 time-steps. For out-distribution evaluation, we sample 10 new environments from the same ranges of parameters, but for parameters not seen during training. We have access to one trajectory from each evaluation environment for adapting the model, and evaluate it on 16 new trajectories.

## C Trajectory based formulation

In practice, $F^e$ is unavailable and we can only approximate it from discretized trajectories. As done in (Kirchmeyer et al., 2022), we use a trajectory-based formulation of Eq. (9). We consider a set of

trajectories discretized over a uniform temporal and spatial grid includes $\frac{T}{\Delta t}(\frac{s}{\Delta s})^{d_s}$ states, where $d_s$ is the spatial dimension. $\Delta t$ and $\Delta s$ represent respectively the temporal and spatial resolution. $T$ and $S$ are the temporal horizon and spatial grid size. Our loss writes as:

$$\mathcal{L}(\theta, \mathcal{D}_{\text{tr}}^e) = \sum_{j=1}^{N} \sum_{k=1}^{(s/\Delta s)} \sum_{l=1}^{T/\Delta T} \|u_j^e(s_k, t_l) - \tilde{u}_j^e(s_k, t_l)\|_2^2 \tag{22}$$

$$\text{where } \tilde{u}^e(t_l) = u_0^e + \int_{t_0}^{t_k} \mathcal{G}_\theta(\tilde{u}^e(\tau))\mathrm{d}\tau$$

$u_j^e(s_k, t_l)$ is the state value in the $j^{th}$ trajectory from environment $e$ at the spatial coordinate $s_k$ and time $t_l \triangleq l\Delta t$. $u^e(t) \triangleq [u(s_1, t), \ldots, u(s_{(S/\Delta s)^{d_s}}, t)]^T$ is the state vector in the $j^{th}$ trajectory from environment $e$ over the spatial domain at time t and $u_0^e$ is the corresponding IC.

# D   Additional results

We conduct a comprehensive ablation study across various datasets to assess the robustness of our method. Our investigation explores several key aspects, including the learnability of PDE parameters within our hybrid framework. We also examine the adaptability of context-based methods to new, unseen environments, with varying numbers of adaptation trajectories $N_{\text{ad}}$. Then, we show the effectiveness of our model in terms of complexity and performance when varying the size of the context vector. Additionally, we study the impact of initialization strategies for adaptation parameters in our meta-learning framework. Finally, we demonstrate the ability of our method to adapt rapidly to unseen environments with minimal gradient updates.

## D.1   PDE parameter estimation

Existing works combining hard physical constraints and data-driven components either learn the PDE parameters or assume availability of the real parameters for a single PDE instance (Yin et al., 2021). In our case, we are trying to learn the PDE parameters in an unsupervised manner for each environment. In section 4.3, we proposed two strategies for estimating the PDE parameters. We report in the Figure 7 the training and adaptation MAE of the PDE parameters pendulum equation, where the first strategy has been adopted. In Figure 10, we report the training and adaptation convergence for the gray-scott equation, where the second strategy has been applied.

Figure 7: MAE loss of the PDE parameters for the Pendulum equation during training and adaptation

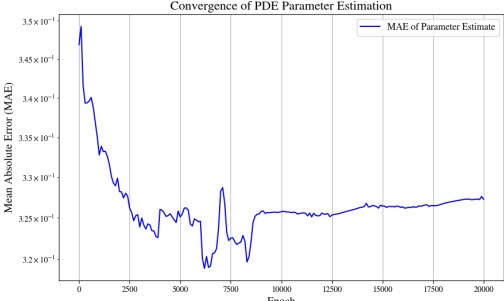

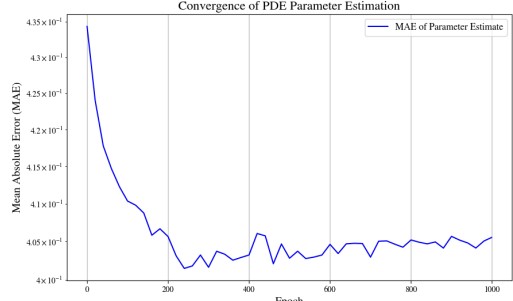

Figure 8: Convergence of PDE parameter estimation during training

Figure 9: Convergence of PDE parameter estimation during adaptation

Our framework successfully estimates PDE parameters during both training and adaptation steps for the Pendulum and Gray-Scott equations. Notably, the accuracy of our framework in predicting PDE parameters is heavily dependent on the initialization of physical parameters. The specific initialization values used for PDE parameters are detailed in Table E.1.

Figure 10: MAE Convergence of the PDE parameters for the Gray-Scott equation

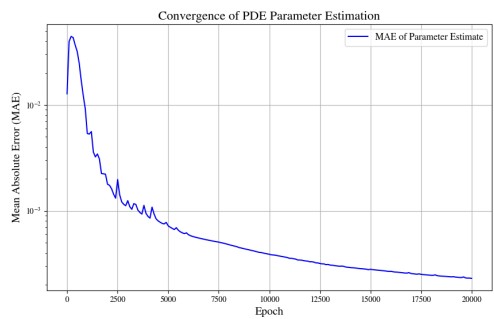

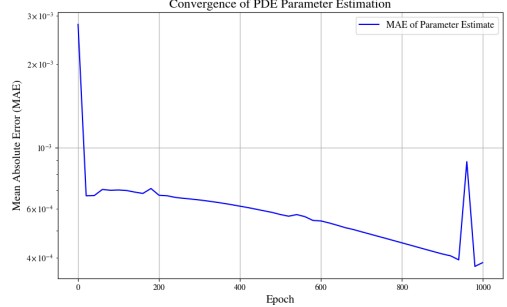

Figure 11: Convergence of PDE parameter estimation during training

Figure 12: Convergence of PDE parameter estimation during adaptation

## D.2 Number of adaptation trajectories

**Context adaptation**   While our methods remain effective on the Gray-Scott and the Burgers equation when adapting to unseen environments, meta-learning frameworks perform poorly on the Kolmogorov flow equations. This is particularly due to (i) the complex dynamics and very different behaviors that appear, (ii) the diversity of behaviors observed within a same environment for different initial conditions. Thus, the number of trajectories for training and for adaptation need to be sufficiently large for learning such dynamics; one-shot learning is therefore a very complex task for such systems. We report in table 5 the results when increasing the number of adaptation trajectories for Kolmogorov flow when adapting only context vectors:

Table 5: Relative MSE loss with respect to number of adaptation trajectories when adapting context vectors $c^e$.

| Model ↓ | Number of adaptation trajectories $N_{\text{ad}}$ | | | |
|---|---|---|---|---|
| | 1 | 4 | 8 | 16 |
| CoDA | 6.23e-1 | 6.22e-1 | 6.22e-1 | 6.20e-1 |
| CAVIA | 6.42e-1 | 6.40e-1 | 6.33e-1 | 6.29e-1 |
| *GEPS* | 5.22e-1 | 5.22e-1 | 5.21e-1 | 5.20e-1 |

We remark that context based frameworks do not scale well when increasing the number of adaptation trajectories.

**Low-rank adaptation**   To overcome this issue, we propose to take advantage of the low-rank adaptation formulation proposed, which enable a cost-efficient adaptation of our model with respect to adapting all the parameters of the model. Instead of adapting $c^e$, we adapt parameters $Ac^eB$. We report results in table 6:

Table 6: Relative MSE loss with respect to number of adaptation trajectories when adapting low rank adaptation parameters.

| Model ↓ | Number of adaptation trajectories $N_{\text{ad}}$ | | | |
|---|---|---|---|---|
| | 1 | 4 | 8 | 16 |
| *GEPS* | 4.5e-1 | 4.23e-1 | 3.97e-1 | 3.64e-1 |

## D.3 Data-driven Parameter initializations

Initialization of adaptation parameters is essential for correctly adapting to environments. We study how initialization impacts the training of our model on the Gray-Scott and Burgers equation. In LoRa,

it is typically advised to initialize $A$ using a Gaussian distribution and set $B = 0$. In table 7, we report the relative loss for different initialization, including LoRa initialization. On both Gray-Scott and Burgers equation, the orthogonal initialization performs best.

Table 7: Relative MSE loss with respect to parameter initialization

| Model initialization | Gray-Scott | Burgers |
|---|---|---|
| Kaiming | 7.26e-2 | 1.50e-2 |
| Xavier | NaN | 1.57e-2 |
| LoRA init | 2.35e-1 | 4.95e-1 |
| Orthogonal | **2.22e-2** | **1.46e-2** |

### D.3.1 Code dimension

In CoDa, the authors advocate to choose relatively small code size, corresponding to the degree of variations defining each environment. In our case, we proposed datasets with more degree of variations than the datasets used by (Kirchmeyer et al., 2022), presenting much more differences, necessitating larger code dimension size. In table 8, we show the performance of our model on both Gray-Scott and Burgers equation when varying the code dimension for CoDA and our method and the increase in number parameters.

Figure 13: Model size with respect to code dimension

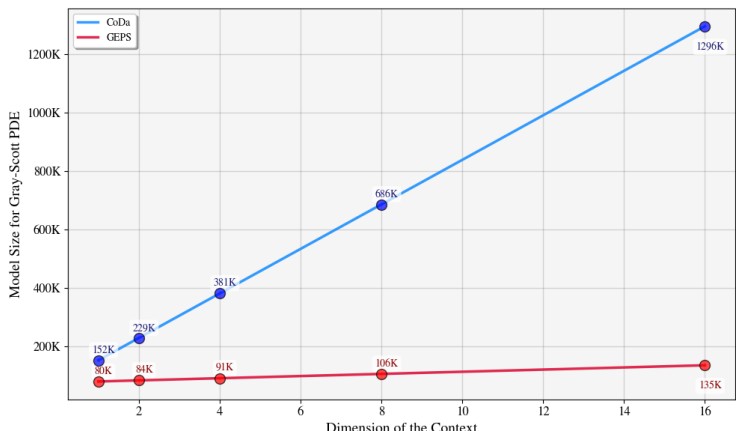

Table 8: Relative MSE on the full trajectory with varying code dimension

| code size ↓ | CoDA | | GEPS | |
|---|---|---|---|---|
| | *Burgers* | *Gray-Scott* | *Burgers* | *Gray-Scott* |
| 1 | 2.04e-2 | 9.72e-2 | 1.49e-2 | 1.10e-1 |
| 2 | 2.03e-2 | 8.73e-2 | 1.35e-2 | 7.12e-2 |
| 4 | 1.96e-2 | 9.29e-2 | 1.46e-2 | **5.65e-2** |
| 8 | 1.78e-2 | 9.38e-2 | 1.43e-2 | 7.27e-2 |
| 16 | 1.84e-2 | 1.01e-1 | **1.30e-2** | 5.82e-2 |

### D.3.2 Adaptation speed

One important property of meta-learning framework is their ability to adapt to unseen environments in few adaptation steps. Our framework allows fast adaptation to new environments, compared to a state-of the art method like CoDA. We report in Figure the convergence of the loss during adaption for both our method and CoDA. With our framework, we are able to adapt to new environments in

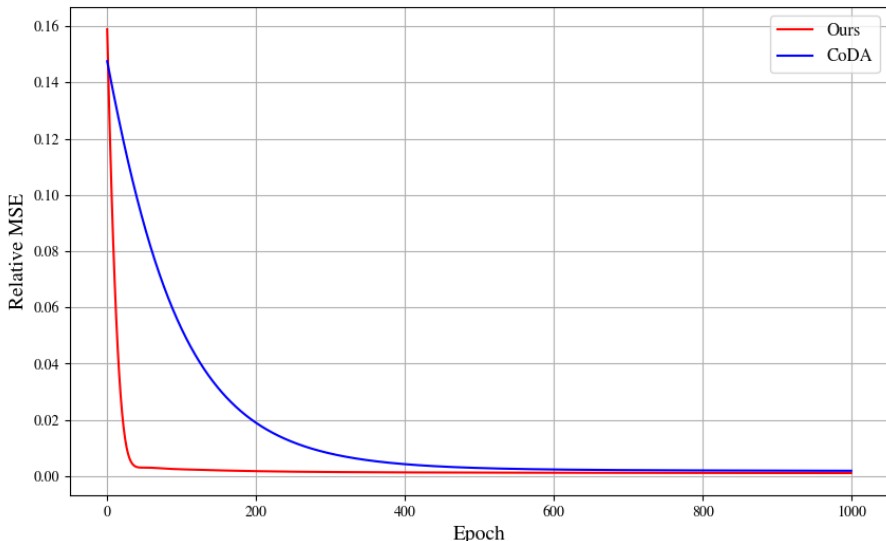

Figure 14: Convergence speed of the model $\mathcal{G}_\theta$ to adapt to new environments for the Burgers dataset

less than 100 steps, compared to CoDA which needs 500 steps. For both runs, we used the same learning rate lr $= 0.01$.

### D.3.3 Out-range temporal horizon generalization

We evaluate the ability of the different adaptation mechanisms for out-range temporal horizon extrapolation. All models have been trained on a temporal horizon $[0, T]$. We thus evaluate the performance of our models when extrapolating outside the trained temporal horizon. For out-range extrapolation, models are evaluated on the horizon $[T, 2T]$. We report in-distribution and out-distribution performance in table 9. Error accumulates over time and leads to lower performance outside the training horizon, but GEPS still outperforms existing baselines.

Table 9: **Generalization results for in-distribution and out-distribution environments** - Test results for out-range time horizon (Out-t). Metric is the relative L2 loss. '−' indicates inference has diverged.

| Type ↓ | Dataset → | *Pendulum* | | *Gray-Scott* | | *Burgers* | | *Kolmo* | |
|---|---|---|---|---|---|---|---|---|---|
| | Model ↓ | *In-d* | *Out-d* | *In-d* | *Out-d* | *In-d* | *Out-d* | *In-d* | *Out-d* |
| Data-driven | LEADS | 6.30e-1 | **7.28e-1** | 1.45e-1 | 1.27e-1 | 4.55e-2 | 3.22e-1 | 9.80e-1 | 1.01 |
| | CAVIA | 40.4 | 1.88 | 3.73e-1 | 5.51e-1 | 6.44e-2 | 6.72e-1 | 1.06 | 1.06 |
| | FOCA | 14.0 | 1.08 | 3.56e-1 | 3.34e-1 | 3.12e-1 | 4.60e-1 | 1.0 | 1.05 |
| | CoDA | 9.54e-1 | 9.75e-1 | 1.52e-1 | 1.19e-1 | 3.41e-2 | 3.35e-1 | 8.29e-1 | 9.99e-1 |
| | *GEPS* | **4.92e-1** | 8.45e-1 | **8.37e-2** | **6.91e-2** | **2.23e-2** | **2.68e-1** | 7.70e-1 | 9.74e-1 |
| Hybrid | APHYNITY | 99.5 | 1.32 | **6.05e-3** | **4.19e-3** | 1.12e-1 | 8.09e-1 | − | − |
| | *GEPS-Phy* | **2.88e-1** | **7.12e-1** | 2.38e-2 | 1.92e-2 | 4.72e-2 | 3.59e-1 | **7.33e-1** | **9.32e-1** |

### D.3.4 Training time and number of parameters

We report in table 10 the training time and number of parameters for each adaptive conditioning method for all datasets:

Table 10: **Number of parameters (# *Params*) and training time (*Time*) for all different adaptive conditioning methods.**

| Method | Pendulum | | Gray-Scott | | Burgers | | Kolmogorov | |
|---|---|---|---|---|---|---|---|---|
| | # *Params* | *Time* | # *Params* | *Time* | # *Params* | *Time* | # *Params* | *Time* |
| LEADS | 43K | 10h | 380K | 14h | 400K | 1d 19h | 1.8M | 2d 23h |
| CAVIA | 9K | 1d 4h | 76K | 1d 15h | 58K | 4d 19h | 278K | 4d 21h |
| CoDA | 43K | 6h | 380K | 5h | 280K | 10h | 726K | 2d 12h |
| *GEPS* | 13K | 5h | 90K | 4h | 69K | 9h | 387K | 2d 2h |

# E    Implementation details

The code has been written in Pytorch (Paszke et al., 2019). All experiments were conducted on a single GPU:NVIDIA RTX A5000 (25 Go). We estimate the computation time needed for development and the different experiments to approximately 200 days.

## E.1    Architecture

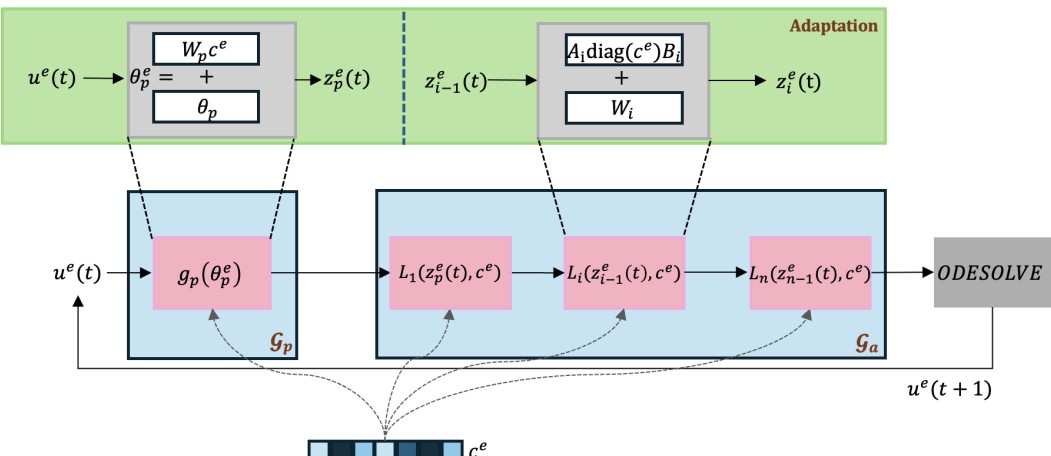

Figure 15: Our adaptation framework for the two instances of our model: agnostic and physics-aware. Blocks in blue refer to the physical model $g_p$ and data-driven component $g_a$. Blocks in pink refer to the trainable modules. The green block describes the adaptation mechanism for the physical and data-driven components. We use a NeuralODE (ODESOLVE) for time-integration.

## E.2    Data-driven model

We implement the dynamical model $\mathcal{G}_\theta$ with the following architectures:

- Pendulum: we used MLPs composed of 4 hidden layers of width 64.

- Gray-Scott: 4-layer 2D ConvNet with 64-channel hidden layers, and $3 \times 3$ convolution kernels with circular padding.

- Burgers: 4-layer 1D ConvNet with 64-channel hidden layers, and $7 \times 7$ convolution kernels with circular padding.

- Kolmogorov flow: 4 Fourier layers with 12 modes and a width of 16.

For all networks, we use Swish activation layers. We use an RK4 solver for the Neural ODE.

### E.3 Physical model

For each dataset, we assume availability of some prior knowledge of the studied dynamical system, given as hard constraints. We report in the table 11 the physical knowledge assumed, the number of training parameters and the values of the parameters used at initialization:

Table 11: **Physical model details**

|  | Physical Model | # trainable params | Parameter initialization |
|---|---|---|---|
| Pendulum | $\frac{d^2\theta}{dt^2} + \omega_0^2 \sin\theta - F\cos(w_f t)$ | 3 | $w_0^2 = 0.1, F = 0.2, w_f = 0.5$ |
| Burgers | $\frac{du}{dt} + \frac{du}{dx} - \nu\frac{d^2u}{dx^2}$ | 1 | $\nu = 1e-2$ |
| Gray-Scott | $\frac{du}{dt} - D_u\Delta u + uv^2 - F(1-u)$ 
 $\frac{dv}{dt} - D_v\Delta v + uv^2 + (F+k)v$ | 2 | $F = 5e-2$ 
 $k = 5e-2$ |
| Kolmogorov | $\frac{\partial\omega}{\partial t} + (\mathbf{u}\cdot\nabla)\omega - \nu\nabla^2\omega$ | 1 | $\nu = 5e-3$ |

### E.4 Optimizer

For all datasets, we use the Adam optimizer and $(\beta_1, \beta_2) = (0.9, 0.999)$ for both training and adaptation. For training, we used a learning rate scheduler which reduces the learning rate when the loss has stopped improving. We set the threshold to $0.01$ with a patience of 250 epochs with respect to the training loss. The minimum learning rate is $1e-5$.

### E.5 Hyper-parameter details

We report the hyper-parameters used for each dataset in the table:

Table 12: **Framework hyper-parameters**

|  | Hyper-parameter | *Pendulum* | *Burgers* | *Gray-Scott* | *Kolmogorov Flow* |
|---|---|---|---|---|---|
| $\mathcal{G}_\theta$ | $c$ | 16 | 4 | 4 | 4 |
|  | depth | 4 | 4 | 4 | 4 |
|  | width | 64 | 64 | 64 | 64 |
|  | activation | Swish | Swish | Swish | Swish |
| Training hyper-parameters | batch size | 8 | 4 | 4 | 4 |
|  | epochs | 20000 | 20000 | 20000 | 20000 |
|  | learning rate | 1e-2 | 1e-2 | 1e-2 | 1e-2 |
|  | Scheduler decay | 0.9 | 0.9 | 0.9 | 0.9 |
|  | Teacher forcing | Yes | No | No | No |
| Adaptation hyper-parameters | batch size | 4 | 4 | 4 | 4 |
|  | epochs | 500 | 500 | 500 | 500 |
|  | learning rate | 1e-2 | 1e-2 | 1e-2 | 1e-2 |

### E.6 Baselines implementation

For all baselines, we followed the recommendations given by the authors.

**ERM experiments** We use the following baselines for the ERM experiments:

- **MP-PDE**: We implement MP-PDE as a 1-step solver, where the time-bundling and pushforward trick do not apply. We use 6 message-passing blocks and $64$ hidden features. We build the graph with the $8$ closest nodes. We use a learning rate of $1e-3$ and a batch size of 16. We train for 5000 epochs on *Burgers* and *Gray-Scott* equations.
- **FNO**: FNO is trained for $5,000$ epochs on *Burgers* and *Gray-Scott* equations with a learning rate of 1e-3. We used 12 modes and a width of 32 and 4 Fourier layers. We also use a step scheduler every 250 epochs with a decay of 0.5.

- **CNN**: We implement a CNN as 1-step solver, composed of 4 layers of 64 features with Swish activation functions. We train our model on 5000 epochs on *Burgers* and *Gray-Scott* equations with a learning rate of $1e-3$. We also use a step scheduler every 250 epochs with a decay of $0.5$.

- **GEPS**: We implement a CNN as 1-step solver with a multi-environment setting, composed of 4 layers of 64 features with Swish activation functions. We train our model on 5000 epochs on *Burgers* and *Gray-Scott* equations with a learning rate of $1e-3$. We also use a step scheduler every 250 epochs with a decay of $0.5$.

**In-distribution and out-distribution experiments**   We use the following baselines for the generalization experiments:

- **LEADS**: We implement *one-per-env* method of the multi-task learning framework LEADS. For all datasets, we train the model during 20000 epochs with a learning rate $lr = 0.01$, with a step scheduler every 250 epochs. We used the same batch size used for our framework across all datasets. All other hyper-parameters are the same than the one used in the paper.

- **CAVIA**: We adapt CAVIA's framework for learning dynamical systems. We train CAVIA during 20000 epochs with a inner learning rate $lr_{inner} = 0.1$ and an outer learning rate $lr_{outer} = 0.001$. For all datasets, we train the model during 20000 epochs with a step scheduler every 250 epochs. We used the same batch size used for our framework across all datasets. We tested different inner step values, and fixed to 7 for all datasets. We followed the authors recommendations for the experiments.

- **FOCA**: We implement a first-order gradient-based method called FOCA to learn dynamical systems. We train CAVIA during 20000 epochs with an inner learning rate $lr_{inner} = 1.0$ and an outer learning rate $lr_{outer} = 0.001$. For all datasets, we train the model during 20000 with a step scheduler every 250 epochs. We used $\tau = 1$ for the exponential moving average and 5 inner steps.

- **CODA**: We implement CoDA, a meta-learning framework for learning dynamical systems. We implemented the l1 and l2 regularization of the CoDA framework. We train the model during 20000 epochs with a step scheduler every 250 epochs. For the size of the contexts, we first fixed the size following the authors recommendations, but we observed improved performance using the same dimension used with our framework.

- **APHYNITY**: We implemented a multi-environment version of the APHYNITY framework. We learn physics for each environment and learn to correct the physics model using a data-driven model that corrects all environments. During adaptation, we tune all the parameters of the model. For all datasets, we train the model during 20000 epochs with a learning rate $lr = 0.001$, with a step scheduler every 250 epochs. We used the same batch size used for our framework across all datasets.

# F   Qualitative results

In this section, we show different visualization of the predictions made by our framework and compare with the different baselines.

## F.1   In-distribution and out-distribution generalization for Burgers dataset

We provide in Figure 16 in-distribution and out-distribution generalization for the Burgers equation.

## F.2   Out-distribution generalization for Gray-Scott dataset

We provide in Figure 17 out-distribution generalization for the Gray-scott equation.

## F.3 In-distribution generalization for Kolmogorov dataset

We provide in Figure 17 in-distribution generalization for the Kolmogorov equation.

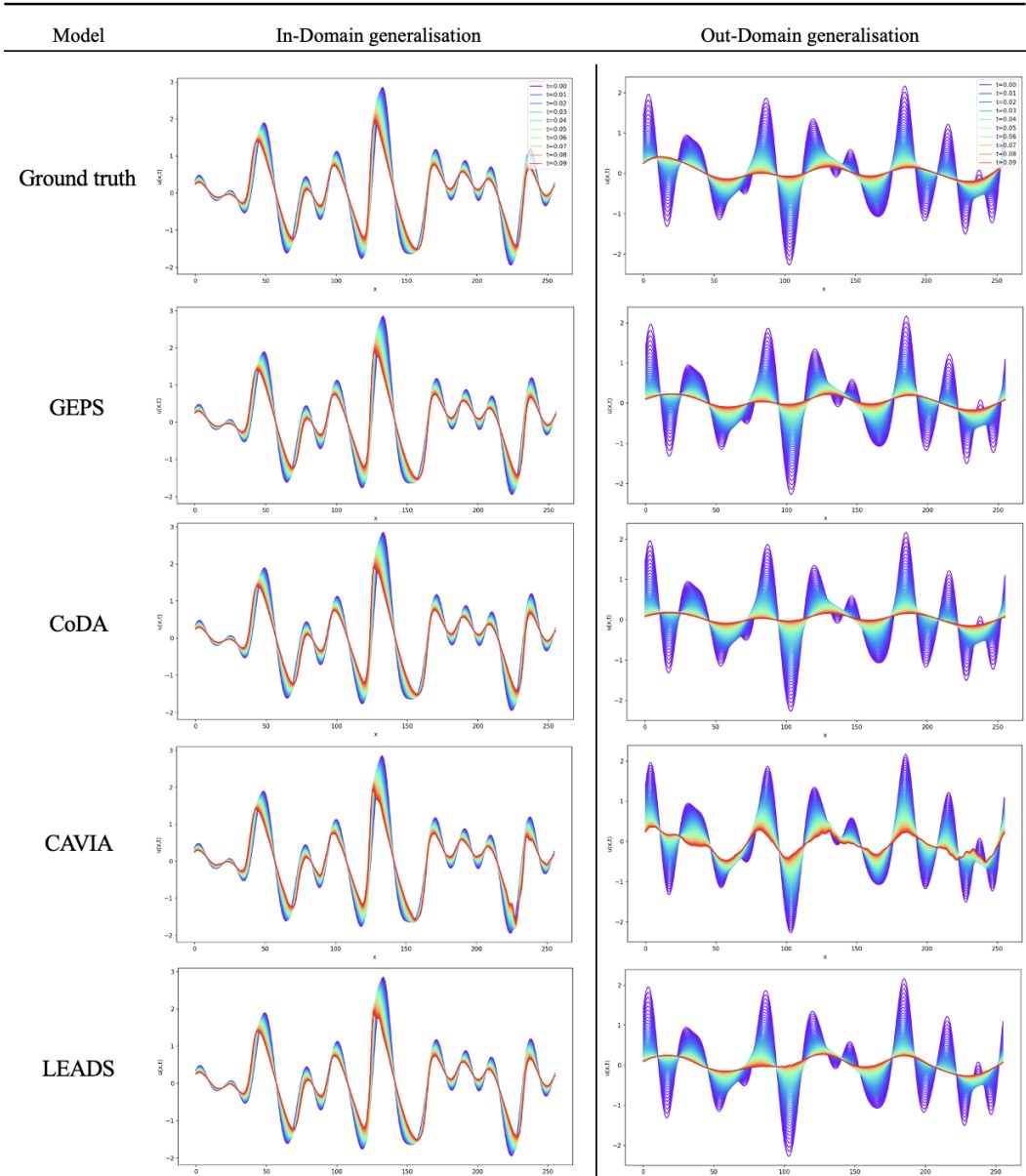

Figure 16: Comparison of in-distribution and out-distribution predictions of a trajectory on 1D Burgers. The trajectories are predicted from t = 0 (purple) to t = 0.1 (red).

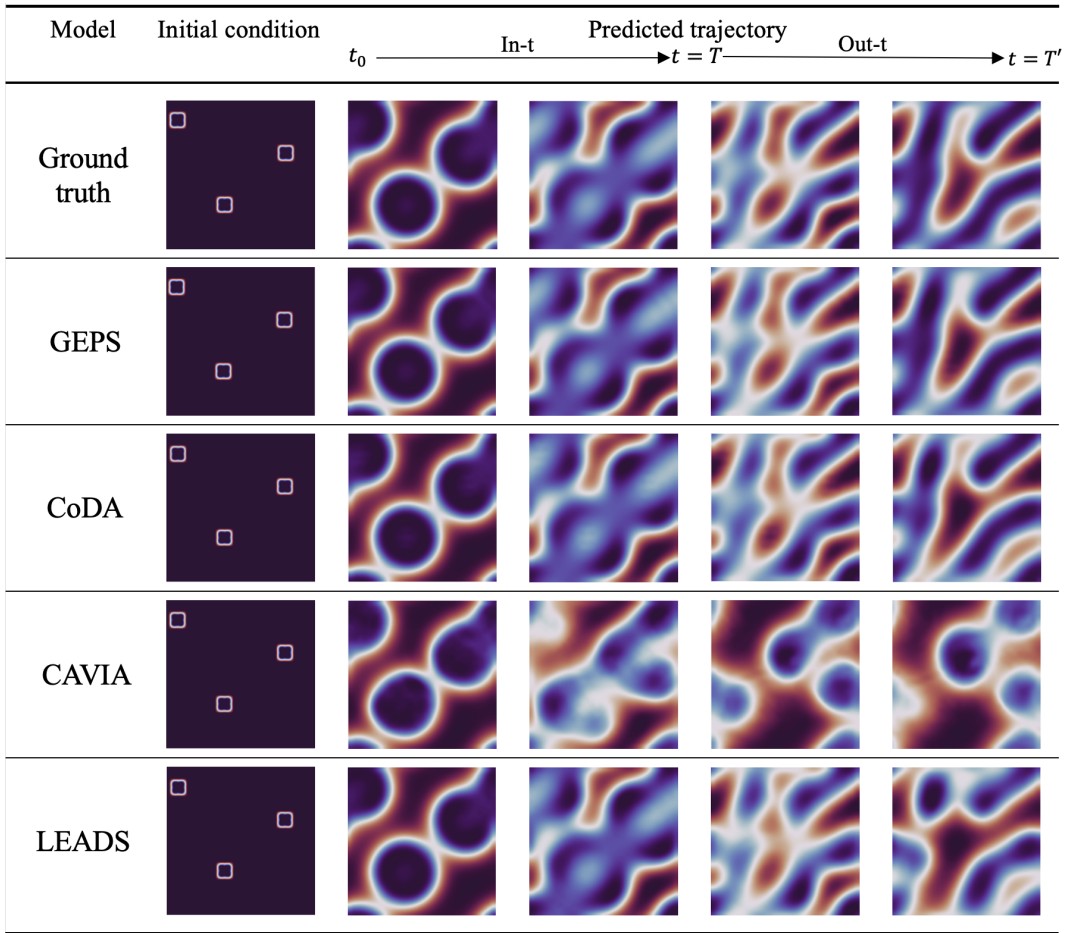

Figure 17: Prediction per frame for our approach on 2D Gray-Scott for an out-of-distribution trajectory. The trajectory is predicted from t = 0 to t = T'. In our setting, T = 19 and T' = 39.

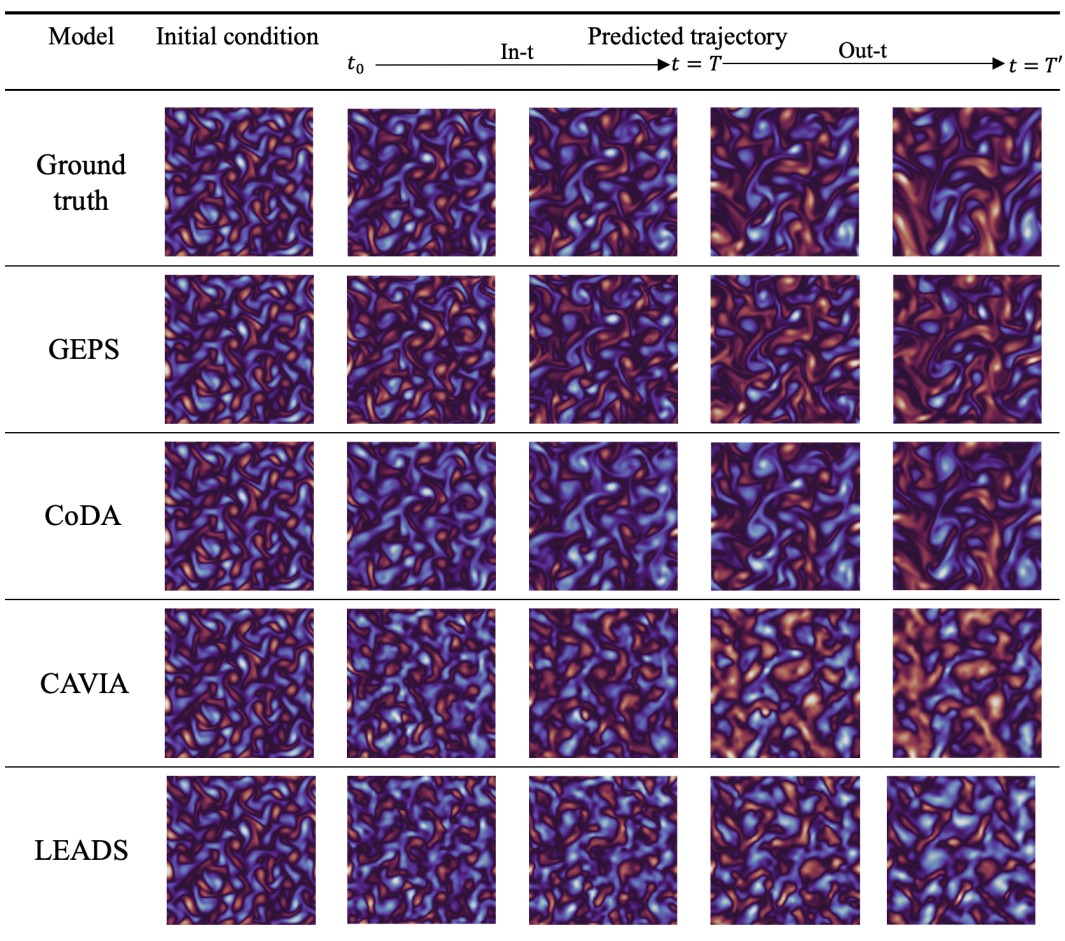

Figure 18: Prediction per frame for our approach on 2D Kolmogorov flow for an in-distribution trajectory. The trajectory is predicted from t = 0 to t = T'. In our setting, T = 19 and T' = 39.

