# OpenReview forum: "Boosting Generalization in Parametric PDE Neural Solvers through Adaptive Conditioning"
_NeurIPS.cc/2024/Conference — NeurIPS 2024 poster_

### Official Review · Reviewer_1xSb · 2024-07-06

**Soundness:** 2
**Presentation:** 2
**Contribution:** 2
**Rating:** 5
**Confidence:** 4

**Summary:**

This paper proposes to solve parametric PDEs with the introduction of context parameters. The low-rank design allows for rapid adaptation to unseen conditions, and the experiments show comparable or slightly better performance of the method than baselines.

**Strengths:**

1. The utilization of context parameters facilitates efficient adaptation to novel environments while minimizing computational overhead.

**Weaknesses:**

1. The main architecture of the proposed model is similar to LoRA, despite the fact that the context parameters are learned jointly during the pre-training stage.  The core innovation lies more in the combination of existing techniques rather than introducing a fundamentally new approach. The motivation of the model design is not well established as well.
2. The experiment results seem not strong, especially for the Gray-Scott equation.
3. The paper briefly mentions the computational efficiency of the proposed method but lacks a detailed analysis.
4. The sensitivity of the proposed method to various hyperparameters is not extensively discussed, such as the size of the context vector and the rank in the low-rank adaptation.

**Questions:**

1. Why Neural ODE is utilized to generate the full trajectory? Do the experiment settings require continuous-time inference?

**Limitations:**

See weaknesses.

---

> ### Author Rebuttal · Authors · 2024-08-07
>
> We're thankful for the reviewer's feedback and have addressed the raised concerns below.
>
> ### The main architecture of the proposed model is similar to LoRA.
>
> The paper primarily focuses on the general adaptation framework rather than the LoRA implementation. The key points are:
>
> - Classical ERM training is inadequate for modeling the variability in dynamical physical processes (framed as parametric PDEs), as demonstrated in section 3.2.
> - An alternative inductive principle is necessary; we propose an adaptive conditioning principle: learning from multiple environments to rapidly adapt to a new one (section 3.3). The core issue is developing an effective meta-learning strategy.
> - Our implementation of this principle is computationally efficient (fast adaptation to new environments) and scalable w.r.t. the number of training samples, via low-rank and first-order optimization.
> - We propose a versatile framework compatible with any NN backbone and demonstrate that physical models can be learned across different environments despite incomplete models and unknown parameters, a novel achievement.
> - It improves the state-of-the-art in learning from multiple environments on new proposed datasets. Experiments show our framework works well with MLP, CNN, and FNO backbones.
>
> ### The motivation of the model design is not well established as well.
>
> The proposed model combines a physics module and an agnostic NN module. The former encodes some prior knowledge available under the form of a PDE that only partly explains the underlying physical phenomenon, while the latter comes as a complement to this partial physical model in order to model the physics not explained by the former.
> Concerning the order of the combination ($g_a \circ g_p$), this order is natural since the NN component $g_a$ comes as a complement to the partial physics encoded in $g_p$.  Alternatives methods exist: [1] propose $g = g_p + g_a$ or [2] proposed a variational formulation. We proposed $g = g_a \circ g_p$, which is more flexible than existing methods, as it does not assume the physical component needed to be completed additive or a probabilistic formulation. Given an initial condition $u_0$, the physical model $g_p$ takes as input $u_0$ and outputs an estimate of the temporal derivative $du/dt$, which is processed by the data driven term $g_a$. The latter then complements the incomplete estimate computed by the physical component.
>
> [1] Yin et al. Augmenting Physical Models with Deep Networks for Complex Dynamics Forecasting, ICLR 2021
>
> [2] Takeishi et al. Physics-Integrated Variational Autoencoders for Robust and Interpretable Generative Modeling, NeurIPS 2021.
>
>
> ### The experiment results seem not strong, especially for the Gray-Scott equation
>
> We have added complementary experiments, including a new state-of-the-art transformer baseline (Transolver) in Figure 1 and Table 2, and a new ablation study, Phys-Adaptor (Tables 3 and 4), which uses a shared NN backbone without adaptation. The Transolver's performance is similar to other backbones, and the ablation highlights the importance of the proposed adaptive framework. Regarding Gray-Scott, we respectfully disagree with your conclusions. Our method outperforms all baselines except APHYNITY in the hybrid setting. Figure 15 qualitatively demonstrates the quality of our predicted trajectories for Gray-Scott.
>
> ### The paper briefly mentions the computational efficiency of the proposed method but lacks a detailed analysis.
>
> Thanks for the suggestion, we have added in the PDF rebuttal page training time and parameters gains for the different multi-environment frameworks, showing the superiority of the GEPS framework.
>
> ### The sensitivity of the proposed method to various hyper-parameters, e.g. context vector size
>
> Again, this might not be sufficiently indicated in the core paper, but for lack of space, we included the ablation analysis in the appendix. More precisely, in the appendix C.3.1, we studied the impact of the context size on the performance and on the number of parameters required for adaptation. The proposed GEPS reaches higher performance at a lower complexity, in terms of parameters, than the reference baseline CODA. Finally in C.2, we studied how the number of adaptation trajectories impact the adaptation performance.
>
> ### Why Neural ODE is utilized to generate the full trajectory? Do the experiment settings require continuous-time inference?
>
> Neural ODE here makes reference to a family of time integration methods. This includes several differentiable PDE solvers. Here we made use of a popular RK4 explicit solver available in the torchdiffeq library. Other alternatives are possible with our framework. For the ERM methods, we used the same time-stepping technique method for GEPS than those used in the ERM methods.
>
> We hope we clarified all your concerns and will make sure everything will be made clearer in the final version.

---

> > ### Comment · Reviewer_1xSb · 2024-08-13
> >
> > Thanks for the authors' responses. My concerns with regard to the efficiency are addressed.
> > However, while I agree with the statement that traditional “ERM” methods may struggle to capture the variability of complex physical systems, especially when only the initial conditions are provided (in which case it would be extremely challenging for the model to make accurate predictions, as different PDE coefficients or other components could lead to divergent future states), the relationship between the motivation and the specific model design is still unclear to me. The results in Appendix C.2 seem to indicate that the original low-rank adaptation method results in stronger performance without introducing much more computational overhead than the method proposed in this paper. Moreover, as suggested by other reviewers, I think that a simple baseline with "shared backbone and domain-specific weights" should be added. Therefore, I will maintain my current score.

---

> ### Author Response · Authors · 2024-08-13
>
> Thank you for taking time to read our answers to your concerns and acknowledging our added results. We would be happy to engage in a further discussion if you allow it.
>
> The motivations served to point out a pitfall in current PDE solvers inductive principle formulation for tackling parametric PDEs. We thus proposed a new inductive principle formulation that could tackle this pointed challenge. As for the specific framework implementation, we aimed to introduce an effective solution applicable to a wide range of neural PDE solvers, supported by two key technical contributions:
> - An effective yet computationally efficient meta-learning formulation, as demonstrated in Table 1 of the rebuttal, which compares favorably with existing frameworks.
> - A new, flexible approach for learning hybrid models within a meta-learning framework, which, to the best of our knowledge, has not been shown before.
>
> Being backbone agnostic and accepting physics if available allow it to handle a large number of PDE solvers encoutered in the literature.
>
> Regarding the simple baseline « shared backbone and domain-specific weights », as mentioned to reviewer m65H, all our baselines are in fact "shared backbone with domain specific-weights", as shown in Figure 6 of [1] in appendix. We think multi-task and meta-learning frameworks, while different in their problem formulation, solve the same optimization problem, as shown in [2].
>
> As per your requests, we added two simple strategies, where we have a shared backbone but have domain specific-weights for the first-layer or the last layer.
>
> We report the results for Gray-Scott and Burgers for out-domain results (similar conclusions have been observed in-domain):
>
> | Model | GS | Burgers |
> |-------|---------|---------|
> | first-layer     |    4.52e-2   |    7.34e-2   |
> | last-layer     |    5.03e-2   |    8.03e-2   |
> | GEPS     |    1.86e-2   |    5.29e-2   |
>
> Concerning the results in Appendix C.2, we actually think this is one particular strength of our method, where we can also tune the low-rank matrices to obtain better accuracy at a reasonable cost if needed. This cost could be significant for large models or for larger number of adaptation samples, justifying why we first proposed to only tune a very small subset of parameters $c^e$.
>
> We hope these results and the already provided experiments answer your concerns. Don’t hesitate to reach out if needed.
>
> [1] Kirchmeyer et al., Generalizing to New Physical Systems via Context-Informed Dynamics Model. ICML 2022
>
> [2] Wang et al., Bridging Multi-Task Learning and Meta-Learning: Towards Efficient Training and Effective Adaptation. ICML 2021

---

> > ### Author Response · Authors · 2024-08-14
> >
> > Dear reviewer,
> >
> > Thank you again for the time taken to review our paper.
> >
> > We would like to kindly remind that we have addressed your last concerns in our last response and hope you can take note of it this before the discussion deadline.
> >
> > If so, we respectfully ask if you could kindly indicate whether your rating has been updated accordingly.
> >
> > Thank you very much for your time, and we look forward to your response.

---

> > > ### Comment · Reviewer_1xSb · 2024-08-14
> > >
> > > Many thanks to the authors for providing additional experiments and explanations. Although I still hold that the technical contribution remains modest, as the original low-rank adaptation method appears to perform comparably to the proposed approach in terms of both efficiency and effectiveness, I will raise my score to 5, as this paper reasonably introduces meta-learning for multi-environment dynamic system modeling.

---

> > > > ### Author Response · Authors · 2024-08-14
> > > >
> > > > We would like to sincerely thank you for your appreciation of our new experiments.
> > > >
> > > > We understand your concern regarding the originality of the technical contributions. We hope that our main message regarding the use of a new inductive principle and the effective solution proposed will still prove valuable to the community in advancing parametric PDE neural solvers.
> > > >
> > > > While our method consistently outperforms all baselines across the datasets, we acknowledge that the improvement in terms of performance is modest. Nonetheless, we believe it is noteworthy that our approach requires 20 times fewer adaptation parameters compared to the best-performing baseline, CoDA. Although this reduction in parameters is manageable for current models, we hope our work encourages the adoption of adaptation mechanisms, especially for foundation models where the computational cost is much greater and efficiency becomes crucial.
> > > >
> > > > Thank you once again for the insightful discussion and for dedicating your time to help us strengthen our paper.

---

### Official Review · Reviewer_m65H · 2024-07-09

**Soundness:** 3
**Presentation:** 2
**Contribution:** 3
**Rating:** 5
**Confidence:** 3

**Summary:**

This paper focuses on the PDE solver generalization and proposes the GEPS model based on a low-rank-based meta-learning strategy. Specifically, the authors config the PDE solver as a low-rank framework, where different environments correspond to domain-specific diagonal matrices. During adaption, only the diagonal matrices are trained. Further, they employ a hybrid framework for incorporating physics prior as a trainable module in the model beginning. The authors provide extensive in- and out-domain experiments and compare the model with diverse baselines.

**Strengths:**

-	This paper focuses on an important problem, which is PDE generalization.
-	The proposed method is reasonable and performs well in most cases.
-	This paper is well-written and clear.

**Weaknesses:**

1.	The technical contribution is quite limited.

Utilizing shared backbone and domain-specific weights for multi-environment or multi-task learning is a widely used design. There are many adaptor-based models, such as [1]. As for the hybrid framework, I think the authors fail to elaborate on the motivation of the overall design. Why use the prior module at the beginning? What about using it at the end of the model or using it as an adaptor? Also, why does the physics module not use the same low-rank design as the data part?

[1] Parameter-efficient Multi-task Fine-tuning for Transformers via Shared Hypernetworks, ACL 2021

In addition, fine-tuning the diagonal matrix is also related to SVD-PINN [2], which should also be included as a baseline.

[2] Svd-pinns: Transfer learning of physics-informed neural networks via singular value decomposition, SSCI 2022

2.	About experiments

Some simple but important baselines are missing, including the adaptor-based method and some generalizable backbones. Specifically, for the adaptor-based method, the authors can train different adaptors for different environments based on a shared backbone. As for the generalizable backbones, they can experiment with advanced neural operators, such as [1,2]. I mean that train a universal model with thes backbones (the PDE coefficients are concatenated with inputs) and directly generalize them to new PDEs or coefficients. These baselines should also be included in Section 3.2 experiments.

[1] Transolver: A Fast Transformer Solver for PDEs on General Geometries, ICML 2024

[2] Scalable Transformer for PDE surrogate modeling, NeurIPS 2023

As a supplement to the main results in Tables 2 and 3, the training time of different methods should be compared.

3.	Comparison with ensemble in Figure 3.

I think ensemble could be a strong baseline, where it performs closely to GEPS in Burgers. Maybe the authors should replace the backbone with more advanced Transformer-based methods.

4.	About the title “adaptive conditioning”.

This paper proposed a low-rank training strategy. Thus, I think the adaptive conditioning is unsuitable.

**Questions:**

About the ERM experiments, will the model know the correct PDE coefficient during training and adaption? Do these experimented ERM baselines use the same information as GEPS?

**Limitations:**

They have discussed limitations.

---

> ### Author Rebuttal · Authors · 2024-08-07
>
> We appreciate the reviewer's thoughtful feedback and recommendations to improve the paper. We have addressed the raised concerns below, including additional experiments as suggested.
>
> ### Utilizing shared backbone and domain-specific weights for multi-environment or multi-task learning is a widely used design.
>
> Maybe the paper was not clear enough, but the main focus is on the general adaptation framework, not the specific implementation. The main messages are as follows:
>
> - Classical ERM training is inadequate for modeling the variability of dynamical physical processes (parametric PDEs), as demonstrated in section 3.2.
> - An alternative inductive principle is needed. We propose an adaptive conditioning principle: learning from multiple environments to adapt rapidly to a new one (section 3.3).
>
> On the technical side, while sharing a backbone and using domain-specific weights is common, current meta-learning methods are costly and do not allow learning PDE solvers on large datasets. Thus, our implementation of this principle:
> -  is computationally efficient (fast adaptation to new environments) and scalable w.r.t. the number of training samples via a low-rank and first-order optimization.
> - is adaptable to any NN backbones. Our experiments show it works well with several backbones (MLP, CNN, FNO).
> - can embed physical models learned on different environments, even with incomplete models and unknown parameters, which has never been shown before.
>
> ### Motivations of the overall hybrid design
>
> The proposed model integrates a physics module $g_p$ encoding prior knowledge from a PDE that only partially explains the physical phenomenon. The NN module $g_a$complements $g_p$ by modeling the unexplained physics.
>
> Regarding the combination order ($g_a \circ g_p$), it is logical that ($g_a$) operates after $g_p$ as it must augment the incomplete knowledge.
>
> Alternative methods exist, such as [1] $g = g_p + g_a$ or [2] a variational formulation. Our approach, $g = g_a \circ g_p$, is more flexible as it doesn't assume an additive or probabilistic formulation. Given an initial condition $u_0$, $g_p$ outputs an estimate of the temporal derivative $du/dt$, which is then refined by $g_a$.
>
> For the implementation, the physical component ($g_p$) is encoded as a differentiable numerical solver. This allows it to be combined with any NN components and the entire architecture to be trained end-to-end. Since the physics component has few free parameters, a low-rank formulation is unnecessary.
>
> [1] Yin et al. Augmenting Physical Models with Deep Networks for Complex Dynamics Forecasting, ICLR 2021
>
> [2] Takeishi et al. Physics-Integrated Variational Autoencoders for
> Robust and Interpretable Generative Modeling, NeurIPS 2021.
>
> ### What about using it as an adaptor?
> We added a new baseline “Phys-Adaptor” in the rebuttal (pdf companion Table 3 and 4) where only $g_p$ act as an adaptor, thus $g_a$ is shared across the environments. The lower performance of this baseline illustrates again the importance of adapting $g_a$.
>
> ### SVD-PINN as a new baseline.
> PINNs methods are data-free, requiring full knowledge of the physical information, which is not the case here. Our framework is applied on two popular settings:
>
> - Pure data-driven approaches: No prior knowledge of the physics is available, and everything must be learned from data.
> - Hybrid formulation: A PDE partially describing the physical system provides some prior knowledge of the physics (section 3.3.3).
>
> These settings correspond to common real-world situations explored in the literature.
>
> ### Adding an adaptor-based method
>
> Our framework can be used with different backbones as indicated above. [3] has been applied on pre-trained transformers and performs adaptation by adding new “adapter” layers, which increase the number of layers. In our setting, we train a model from scratch to learn to be adapted to new environments efficiently. During adaptation, all baselines keep the same nb. of layers and only tune a small set of parameters, e.g. $c^e$.
>
> [3] Parameter-efficient Multi-task Fine-tuning for Transformers via Shared Hypernetworks, ACL 2021
>
> ### Experiments with advanced neural operators.
>
> We added the **Transolver** baseline in our ERM experiments in the rebuttal (Fig 1 and Table 2). The performance is similar to other ERM baselines, verifying that an adaptation framework is necessary regardless of the backbone used.
>
> ### Adding training times for the different methods
>
> We have added the training times and the number of parameters for each method in the PDF rebuttal page (table 1), showing that GEPS shows a huge time gain compared to gradient-based methods (CAVIA baseline) and that our method is less costly in terms of parameters compared to the reference baselines CoDA and LEADS.
>
> ### Ensemble baseline with a Transformer-based method.
>
> For the ensemble method, we used the same model (CNN) as in our GEPS framework for a fair comparison. Besides, the above experiments with the Transolver shows that the backbone itself does not make a significant difference.
>
> ### The "adaptive conditioning" term is unsuitable.
>
> We hope to have clarified this issue. The paper main message is that ERM inductive principle alone is not sufficient for modeling multiple complex physical dynamics and that another inductive principle should be used. Hence the adaptation inductive principle developed in the paper.
>
> ### Knowledge of the PDE coefficients for ERM experiments
> In all our experiments, the PDE coefficients are unknown.
>
> ### Same knowledge for ERM methods and GEPS?
> The assumptions are the same for all the methods (baselines and GEPS). The only difference is in the induction mechanism: ERM methods assume that all data come from the same distribution, while GEPS assumes that data come from different distributions and incorporates an adaptation mechanism to adapt the network to each distribution.
>
> We appreciate your feedback and hope we answered your concerns.

---

> > ### Comment · Reviewer_m65H · 2024-08-13
> >
> > Thanks for your response and new results. The experiments about Transformer-based models and efficiency helped me understand this method better.
> >
> > However, I still think the "shared backbone and domain-specific weights" setting is a strong and necessary baseline. Note that this baseline is not about meta-learning, it is just a classical multitask learning backbone. Besides, since the authors state that "adaptive conditioning" refers to "learning from multiple environments to adapt rapidly to a new one", the above "shared backbone and domain-specific weights" can also be viewed as a special "adaptive conditioning", where the domain-specific head can "adaptively" utilize the shared representations.
> >
> > Thus, I decided to keep my original score.

---

> ### Author Response · Authors · 2024-08-13
>
> Thank you for your review and for appreciating our new experiments. We would like to engage in a further discussion if you allow it.
>
> We agree that having a « shared backbone and domain specific weights » setting is an important baseline. However, as recent papers suggest [1], meta and multi-task learning, while conceptually different in the problem formulation, share the same optimization problem in reality.
>
> All our baselines, whether meta-learning or multi-task learning, do have a shared backbone and domain specific weights, as you are suggesting. In CoDA [2], the paper provide a good analysis of the differences between all the different multi-environment learning frameworks using a shared backbone with domain specific weights, in Figure 6 in their supplementary material.
>
> However, as per your request, we added two different baselines.
>
> We proposed two different strategies:
> - having a shared backbone but having domain specific weights for the first layer
> - having a shared backbone but having domain specific weights for the last layer
>
> We report the results for Gray-Scott and Burgers for out-domain results (similar conclusions have been observed in-domain):
>
> | Model | GS | Burgers |
> |-------|---------|---------|
> | first-layer     |    4.52e-2   |    7.34e-2   |
> | last-layer     |    5.03e-2   |    8.03e-2   |
> | GEPS     |    1.86e-2   |    5.29e-2   |
>
> We hope this and the already provided new experiments answer your concerns. We are happy to discuss more about it if needed.
>
> [1] Wang et al., Bridging Multi-Task Learning and Meta-Learning: Towards Efficient Training and Effective Adaptation. ICML 2021
>
> [2] Kirchmeyer et al., Generalizing to New Physical Systems via Context-Informed Dynamics Model. ICML 2022

---

> ### Comment · Area_Chair_mvhZ · 2024-08-13
>
> Dear Reviewer,
>
> Please have a look at the last part of the author's rebuttal, the other reviews and indicate if your rating has been updated based on that.
>
> Your Area Chair.

---

> ### Author Response · Authors · 2024-08-14
>
> Dear reviewer,
>
> As the discussion period comes to an end, we would like to express our gratitude once again for your time and for highlighting important concerns that we hope have been addressed.
>
> You mentioned the absence of a baseline involving "a sharing backbone with domain-specific weights." We would like to kindly remind that we have addressed this concern in our last response and hope you can take note of it this before the discussion deadline.
>
> If so, we respectfully ask if you could kindly indicate whether your rating has been updated accordingly.
>
> Thank you very much for your time, and we look forward to your response.

---

> > ### Comment · Reviewer_m65H · 2024-08-14
> >
> > I would like to thank your effort in providing new experiment results. It resolves my concern about baselines.
> >
> > I still hold concerns about "adaptive conditioning". In your definition, this concept is too wide to accurately describe your method. Also, retraining the model to fit a new setting is not "adaptive". I think the presentation of this paper needs a major revision.
> >
> > However, I appreciate your effort in providing new experiments and adding new baselines. I will raise my score to 5.

---

> ### Author Response · Authors · 2024-08-14
>
> We appreciate that our new experiments have addressed your concerns. We would also like to clarify the point you raised:
>
> **Retraining the model to fit a new setting is not adaptive.**
>
> We understand that the term "adaptive conditioning" might be confusing. In the meta-learning community, "adaptation" refers to the process where a model, trained on a range of environments, is efficiently retrained when exposed to data from a new distribution (or environment) to adjust to that new context.
>
> This is the concept we aimed to convey in our title: rather than performing direct inference on data from new environments (which underperforms compared to our approach), parametric neural PDE solvers should incorporate an adaptation mechanism. This mechanism enables the model to quickly learn from minimal data in these new environments, thereby enhancing its generalization capabilities.
>
> **This concept is too broad to accurately describe your method.**
>
> We recognize the concern about the breadth of "adaptation." Our paper primarily highlights a limitation in existing methods for solving parametric PDEs and demonstrates that an adaptive mechanism can address this issue, independent of the specific method proposed. While we provide a concrete solution, our main contribution is more general.
>
> However, if you believe the title should be more specific to our method, we can revise it to: Boosting Generalization in Parametric PDE Neural Solvers through Context-Aware Adaptation.
>
> Thank you for the valuable feedback and the constructive discussion, which have helped us strengthen the quality of our paper.

---

### Official Review · Reviewer_JETf · 2024-07-12

**Soundness:** 3
**Presentation:** 3
**Contribution:** 2
**Rating:** 5
**Confidence:** 2

**Summary:**

In this paper, we propose a meta-learning method called GEPS, which utilizes an adaptation approach to generalize a PDE solver to unseen environments. This method demonstrates better generalization compared to classical ERM approaches.

**Strengths:**

This model can adapt to a new environment \( f^e \) in one shot and predict the time trajectory of the corresponding PDE at inference time. Specifically, the proposed model outperforms other existing meta-learning methods both in in-domain and out-domain environments of the initial conditions, as well as in in-range and out-of-range (extrapolation) time horizons.

**Weaknesses:**

Section 3.2 provides the motivation for the proposed method, but I am curious about how this motivation relates to the methods proposed in Sections 3.3.2 and 3.3.3. It would be helpful to explain how the issues identified with existing methods in the motivation section led to the use of neural ODEs and the model structure presented in Figure 4. Additionally, considering the results in Tables 2 and 3, the improvement in error scale compared to existing meta-learning methods does not seem significant. Figures 14 and 15 also show minimal differences. Therefore, it would be beneficial to experimentally demonstrate other advantages or unique aspects of the proposed GEPS method beyond its accuracy.

**Questions:**

* The overall relative error scale of FNO and MPPDE in Figures 2 and 3 and Table 1 is around 1e-1 (even when the environment and trajectory are sufficient), which appears to be generally larger than the errors presented in the respective original papers. Is this due to the difficulty of the equations, or is there another reason for the relatively large errors?

* I am not quite clear on how the physical knowledge insertion $g_p$ and the data augmentation $g_a$ in Equation (6) are each encoded. Could you provide a more detailed explanation of the encoding methods?

* What is the significance of the order of $g_p$ and $g_a$? Would it cause problems if $g_a$ were applied before $g_p$?

**Limitations:**

See above comments.

---

> ### Author Rebuttal · Authors · 2024-08-07
>
> We're thankful for the reviewer's helpful feedback and have addressed the raised concerns below.
>
> ### Section 3.2 provides the motivation for the proposed method, but I am curious about how this motivation relates to the methods proposed in Sections 3.3.2 and 3.3.3:
>
> The main messages are:
>
> - Classical ERM training is inadequate for modeling the variability in dynamical physical processes (framed as parametric PDEs), as shown in section 3.2.
> - An alternative inductive principle is needed. We propose an adaptive conditioning principle: learning from multiple environments to rapidly adapt to a new one (section 3.3). The key focus is developing an effective meta-learning strategy.
> - We propose an implementation that is computationally efficient (fast adaptation to new environments) and scalable w.r.t. the number of training samples via low-rank (3.3.1) adaptation and first-order optimization (3.3.2). This framework is adaptable to different NN backbones. Experiments show it works well with MLP, CNN, and FNO.
>
> To further assess the generality and applicability of our framework, we consider two settings:
> - Pure data-driven approaches with no prior physics, where everything is learned from data.
> - Hybrid formulations with a PDE prior that partially describes the underlying phenomenon (section 3.3.3).
>
> These settings correspond to common real-world situations explored in the literature.
>
> **Use of Neural ODE**: "Neural ODE" refers to a family of time integration methods. We use a popular RK4 solver, but other solvers could also be used. This is only a relevant technical choice, e.g., for the ERM methods, we used the same time-stepping method proposed in the corresponding papers (FNO, MPPDE, Transolver, CNN) for GEPS.
>
> ### The improvement in error scale compared to existing meta-learning methods does not seem significant. Figures 14 and 15 also show minimal differences.
>
> We have now provided error bars for in-domain and out-domain experiments (tables 3 and 4 in the pdf companion), highlighting that the results obtained are consistent and significant for most cases. Regarding qualitative results, all the methods are able to reconstruct Burgers trajectory (Fig 14). For Gray-Scott (Fig 15), GEPS clearly outperforms CAVIA and LEADS. Only CoDA matches GEPS, but diverges from the ground-truth for out-range time horizon. For the Kolmogorov dataset (Fig 16), only GEPS is able to predict accurately the trajectory, and starts to diverge from ground truth for out range time horizon.
>
> ### Advantages or unique aspects of the proposed GEPS method beyond its accuracy.
>
> - We provided training times and number of parameters of GEPS compared to other methods in the PDF rebuttal page (table 1): GEPS is faster to train and requires less parameters than concurrent approaches.
> - in appendix C.2, we showed that unlike other methods, GEPS performance increase with the number of adaptation trajectories while other methods performance quickly saturate.
> - in appendix C.3.1, we showed that GEPS achieves greater performance and scales better in terms of trainable parameters compared to reference CoDA when increasing the context vector size.
> - In C.3.2, we experimentally observed that GEPS can adapt to new environments faster than CoDA.
>
> ### The overall relative error scale of FNO and MPPDE in Figures 2 and 3 and Table 1 is around 1e-1 which appears to be generally larger than the errors presented in the respective original papers. Is this due to the difficulty of the equations, or is there another reason for the relatively large errors?
>
> The error of FNO and MP-PDE is higher because the datasets used in the respective papers have been generated by a single equation with fixed parameter values while we consider multiple environments (i.e. a distribution of parameters).
> Whatever the backbone, including FNO, MP-PDE and the new Transolver (see additional experiments below), the ERM induction principle is not adequate for learning from multiple physics environments when only an initial condition is given (Initial value problems).
>
> Note that we have added a SOTA Transformer neural operator (NO) the **Transolver**, to the experimental comparison (see Fig. 1 and table 2 in the in the pdf companion), with the same conclusions.
>
> ### How the physical knowledge $g_p$ and the data augmentation $g_a$ in Equation (6) are each encoded?
>
> The physical component of the model is encoded as a differentiable numerical solver. Being differentiable, the latter can be combined with any NN components and the whole architecture trained end to end. As for the combination in equation 6, given an initial condition $u_0$, the physical model $g_p$ takes as input $u_0$ and outputs an estimate of the temporal derivative $du/dt$, which is processed by the data driven term $g_a$. The latter then complements the incomplete estimate computed by the physical component.
>
> ### What is the significance of the order of $g_p$ and $g_a$ Would it cause problems if $g_a$ were applied before $g_p$
>
> Different approaches have been considered in the literature. [1] considered that the data term and the physical term should be combined in an additive manner: $g = g_a + g_p$. In our case, we think that $g = g_a \circ g_p$ is a more general formulation. As for the order, since one makes the assumption that $g_p$ can be incomplete, it is natural to consider that $g_a$ augments this incomplete knowledge and operates after the physical component.
>
> [1] Yin et al. Augmenting Physical Models with Deep Networks for Complex Dynamics Forecasting, ICLR 2021
>
> We hope it answers your remarks. We will make sure that yours concerns are made clearer in the final version.

---

> > ### Comment · Reviewer_JETf · 2024-08-12
> >
> > Thank you for taking the time to explain my questions in detail and for showing the revised figures and tables. While I understand your explanation, I still find the motivation for the proposed method unclear (as pointed out by other reviewers regarding the lack of technical contributions). Additionally, in Tables 3 and 4, it seems there isn't a significant accuracy improvement compared to other benchmark methods, even with the error bars added from multiple experiments. Therefore, I will maintain my weak accept score of 5. However, I believe that the motivation for the proposed method and the advantages or unique aspects of the proposed GEPS method, which you explained to me, should be added and emphasized more in the revised version of the paper. Thank you.

---

> ### Author Response · Authors · 2024-08-12
>
> Thank you for your thoughtful review and for recognizing the additional experiments we included in response to your concerns.
>
> We would like to emphasize again that, as indicated, our main contribution lies not solely in the technical advancements, but in highlighting the inadequacy of classical ERM for parametric PDEs and proposing an alternative inductive principle. We believe that this change in perspective represents a significant step forward that could impact recent trends in foundation models for PDEs, for example, those aimed at solving parametric PDEs.
>
> In addition to this conceptual advancement, we introduced an effective solution applicable to a wide range of neural PDE solvers, supported by two key technical contributions:
>
> - An effective yet computationally efficient meta-learning formulation, as demonstrated in Table 1 of the rebuttal, which compares favorably with existing frameworks.
> - A new, flexible approach for learning hybrid models within a meta-learning framework, which, to the best of our knowledge, has not been shown before.
>
> Respectfully, we would like to note that a score of 5 typically indicates a borderline acceptance, not a weak one.
>
> We will incorporate your feedback into the revised version. Thank you again for your valuable insights, and we are happy to engage in further discussion.

---

### Official Review · Reviewer_3TLD · 2024-07-13

**Soundness:** 3
**Presentation:** 3
**Contribution:** 2
**Rating:** 6
**Confidence:** 3

**Summary:**

The authors propose GEPS, a method for meta-learning neural solvers for parametric PDEs. Similarly to gradient-based meta-learning methods such as MAML, the authors formulate the problem as a two-step optimization problem: the goal of the first step is to learn a model that's able to easily adapt to new environments via learning context-dependent parameters in the second step. A data-driven and physics-based decomposition is also proposed. The authors evaluate on four canonical dynamical systems and demonstrate state-of-the-art performance.

**Strengths:**

- The paper is well-written and clear.
- The proposed method is straightforward and effective. The low-rank parameterization applied to meta-learning for dynamical systems is novel, as far as I am aware.
- Experiments are thorough and comparisons to other meta-learning models are detailed. Evaluated on four canonical dynamical systems, the results are compelling.

**Weaknesses:**

- More experiments investigating the physics-aware component of the model would be illluminating. It's mentioned in Section 3.3.3 for the method to incorporate incomplete physics information, e.g. incomplete set of terms or inexact coefficients. However, evaluations of the hybrid method are only done with full knowledge of the system coefficients.
- More ablation studies could be useful to further clarify the importance of each component of the proposed model. For example, it's unclear to what extent the improved performance is due to the specific low-rank parameterization (Eqn 3).
- It would also be helpful to include the number of parameters of all methods evaluated.

**Questions:**

- Why is it that for Burgers, the hybrid setting of GEPS performs worse than the purely data-driven setting? Does this suggest that the data-driven/physics-based decomposition (Eqn 6) is not expressive enough to handle certain PDEs?
- In Section 3.3.3, two strategies for adapting PDE parameters to a new environment are proposed (lines 213-219): learning $c^e$ only or learning both $c^e$ and $\theta^e_p$. Which strategy is used for Tables 2 and 3?

**Limitations:**

The authors address the limitations.

---

> ### Author Rebuttal · Authors · 2024-08-07
>
> We appreciate the reviewer's thoughtful feedback and have addressed the raised concerns below.
>
> ### More experiments investigating the physics-aware component of the model would be illuminating:
>
> For the hybrid physics-ML setting, we make two assumptions:
> - the physics is only partly known and shall be completed by a NN component for the targeted forecasting task; this is the forward problem. For the pendulum, the physics component ignores the damping term,  for Burgers and Kolmogorov, the model ignores both the forcing and the LES terms. The only exception is the Gray-Scott example, for which one assumes full knowledge of the equation, but the physical parameters are unknown and shall be estimated (see below).
>
> - for all datasets, the parameters of the partial physics component are unknown and shall be estimated - this is the inverse problem. The list of parameters to be estimated is provided in appendix D.3 - table 9. For example for the pendulum, 3 parameters $(\omega_0, F, w_f)$ are estimated. Figures 7 to 11, appendix C1, show the error (MAE) for the estimation of the parameters for pendulum and Gray-Scott (this is an average over all the parameters - 3 for the pendulum, 2 for GS).
>
>
> ### More ablation studies could be useful to further clarify the importance of each component of the proposed model:
>
> We report 4 different ablations that could justify the importance of our low-rank module.
>
> We have added a new baseline denoted **Phys-adaptor**, where we only adapt the physical component to new environments while the data-driven component is shared across all environments, without (low-rank) adaptation. The results, indicated on tables 3 and 4 on the pdf file - row **Phys-adaptor**, show that this model performs worse over all the datasets compared to GEPS, highlighting the importance of adaptation for the NN component.
>
> Other ablations appear in the appendix.
> - in section C.2, we evaluated different context-based meta-learning frameworks (CAVIA, CODA, GEPS) during adaptation while increasing the number of samples per environment. We demonstrated that low-rank adaptation performs better than the alternatives for complex PDEs (Kolmogorov). For example, its performance improves with the number of training trajectories, whereas existing methods quickly reach a plateau and do not show further improvement.
> - in section C.3.1, we showed that our framework scales better than reference method CoDA in terms of numbers parameters and performance when varying the size of the context.
> - Figure 13, appendix C.3.2, shows that our method allows faster convergence during adaptation compared to reference CoDA.
>
> ### It would also be helpful to include the number of parameters of all methods evaluated.
>
> We reported in the PDF rebuttal page (table 1) the training time and the number of parameters used for each baseline for 1D (Burgers) and 2D (Gray-Scott) PDEs.
>
> ### Why is it that for Burgers, the hybrid setting of GEPS performs worse than the purely data-driven setting?
>
> We were also surprised by this results, and redo the experiments. We have no clear explanation, but we believe that the high flexibility of our formulation for combining $g_p$ and $g_a$ can make training complex for some PDEs and causes over-fitting, especially considering the low-amount of data considered in our setting.
>
> ### Two strategies for adapting PDE parameters to a new environment are proposed. Which strategy is used for Tables 2 and 3 in the paper?
>
> For Pendulum and the Burgers, we used the context $c^e$ to learn the physical parameters. For Gray-Scott and the Kolmogorov, we directly learned the physical parameters without using the context vectors. The two strategies provide similar results, we reported in the paper the best results among the two alternatives.
>
> We hope we answered the main concerns which will be made clearer in the final version of the paper.

---

> > ### Comment · Reviewer_3TLD · 2024-08-12
> >
> > Thanks to the authors for the detailed response and clarifications. I believe inclusion of these additional details will strengthen the paper. I will keep my score as is.

---

> > > ### Author Response · Authors · 2024-08-13
> > >
> > > We appreciate your positive feedback. Please don't hesitate to reach out if you have further suggestions or questions.

---

### Author Rebuttal · Authors · 2024-08-06

We thank all the reviewers for their comments and suggestions. We are encouraged that they found our method clear and well-written (Reviewer 3TLD, Reviewer m65H). We particularly appreciate that you found we tackle an important challenge (Reviewer m65H) and provide a straightforward yet strong and computationally effective solution (Reviewer 3TLD, Reviewer 1xSb), well evaluated on a variety of datasets against different baselines, showing SOTA or competitive performance (Reviewer 3TLD, Reviewer JETf, Reviewer m65H). We carefully answered all the concerns raised by each reviewer. We address in this section main concerns shared by the reviewers and we introduce additional experiments following reviewers recommendations.

## Additional experimental results - provided in the pdf companion one page:

- Computational efficiency (All reviewers): Table 1 reports training time and number of parameters for best multi-environment frameworks for Burgers and Gray-Scott equations.

- Error bars (Reviewer JETf): we report error bars for in-range time horizon (out-range horizon has been removed due to lack of space) for both in-domain (Table 3) and out-domain (Table 4) experiments, showing that our results are consistent and significant, training all different frameworks on 3 different seeds.

- Physical adaptor baseline (Reviewer 3TLD, Reviewer m65H): we propose a new baseline “Phys-Adaptor” in table 3 and 4 following reviewer m65H recommendation where only the physical component is adapted and the neural network is shared across all environments. The lower performance of this setting demonstrates the importance of our low-rank adaptation mechanism (Reviewer 3TLD).

- **Transolver** baseline for ERM methods (Reviewer m65H): we have added an advanced Transformer model to our ERM baselines following Reviewer m65H  recommendation for both IVPs in Fig. 1 (Burgers has been removed for lack of space, but similar results have been observed) and when considering a historical window of past states as input (Tab 2), confirming our observations that whatever  the NN backbone used, ERM methods are not effective for multi-environments datasets.

## Main concerns

### Novelty of the adaptation mechanism (Reviewer m65H, Reviewer 1xSb):
We want to highlight that the focus of the paper is on introducing a general adaptation framework suitable for modeling complex PDEs, and not on the implementation of a specific NN model as some of the remarks suggest.  The main messages are:

- Classical ERM training is inadequate for modeling the variability in dynamical physical processes (framed as parametric PDEs), as shown in section 3.2.
- As an alternative, we propose an adaptive conditioning principle: learning from multiple environments to rapidly adapt to a new one (section 3.3). The key focus is developing an effective meta-learning strategy.
- Our implementation of this principle is computationally efficient (fast adaptation to new environments) and scalable w.r.t. number of training samples via a low-rank and first-order optimization formulation. We demonstrate that this model improves the state-of-the-art for learning from multiple environments.
- This framework is adaptable to different NN backbones. Experiments show that it works well with MLP, CNN, and FNO. We also demonstrate that physical models can be embedded and learned across different environments, even when incomplete models and with unknown parameters, which has not been shown before.
- We have validated the method on three new datasets, with multiple environments generated using a large diversity of parameters such as  pde coefficients, domain definition or forcing terms.

### Model design choice (Reviewer JETf, Reviewer m65H):
- For the hybrid modeling setting, the proposed model integrates a physics module and an agnostic NN module. The physics module encodes prior knowledge from a PDE that only partially explains the underlying physical phenomenon, while the NN module complements this by modeling the unexplained physics.

- Regarding the combination of the physical and data-driven modules ($g_a \circ g_p$), this order is logical because the NN component ($g_a$) complements the partial physics encoded in ($g_p$). Alternative methods exist, such as [1] $g = g_p + g_a$ or [2] a variational formulation. We propose $g = g_a \circ g_p$, which is more flexible as it doesn't assume an additive or probabilistic completion of the physical component.

- For time integration, we considered NeuralODE, referencing a family of time-stepping methods. In our implementation, we used a differentiable RK4 numerical solver, though other solvers could also be used. For a fair comparison with ERM methods, we used the same time stepping method proposed in the corresponding reference papers for GEPS.

We acknowledge that Figure 4 in the paper might imply our framework works only with NeuralODE. We will clarify this in the final version.

[1] Yin et al. Augmenting Physical Models with Deep Networks for Complex Dynamics Forecasting, ICLR 2021

[2] Takeishi et al. Physics-Integrated Variational Autoencoders for Robust and Interpretable Generative Modeling, NeurIPS 2021.

### Computational efficiency of our method (All reviewers)

- We have added the training times and number of parameters for all the methods in table 1 for the Burgers and Gray-Scott equations. We observe that GEPS is more efficient in terms of training time and in terms of parameter gains.
- In the main paper (c.f. Fig. 12 and Table 8 in the appendix), we showed that our method performs and scales better in terms of accuracy and total training parameters when varying the context vector size compared to reference CoDA.
- Also, in Fig 13. in appendix, we observe that our method adapts faster (less steps) to new environments than CoDA.

We hope this clarifies the main concerns pointed out by reviewers and will make sure that the different points discussed and new experiments will be added in the final version.

---

### Comment · Area_Chair_mvhZ · 2024-08-09

The authors-reviewers discussion period has now started.

@Reviewers: Please read the authors' response, ask any further questions you may have or at least acknowledge that you have read the response. Consider updating your review and your score when appropriate. Please try to limit borderline cases (scores 4 or 5) to a minimum. Ponder whether the community would benefit from the paper being published, in which case you should lean towards accepting it. If you believe the paper is not ready in its current form or won't be ready after the minor revisions proposed by the authors, then lean towards rejection.

@Authors: Please keep your answers as clear and concise as possible.

The AC

---

### Decision · Program_Chairs · 2024-09-25

**Decision:**

Accept (poster)

**Comment:**

The paper proposes a novel method, GEPS, to improve the generalization of neural PDE solvers to unseen dynamics. GEPS employs a low-rank meta-learning strategy to enable efficient adaptation to new environments by adjusting only a small set of context parameters. The method incorporates physics knowledge through a hybrid framework, combining a physics module with a data-driven module. Extensive experiments demonstrate the effectiveness of GEPS in various spatio-temporal forecasting tasks.

Strengths:

The paper is easy to follow, with clear explanations and concise language.
The low-rank meta-learning strategy for adaptive conditioning is a novel contribution to the field of neural PDE solvers.
The authors provide extensive experiments to validate the effectiveness of GEPS, demonstrating its superiority over existing methods in many cases.
The incorporation of physics knowledge through a hybrid framework offers a flexible approach to modeling complex PDEs.

Weaknesses:

While the low-rank meta-learning strategy is novel, the overall approach of using a hybrid framework and domain-specific weights is not entirely new.
The paper could provide a more detailed explanation of the motivation behind the specific design choices, such as the use of a prior module.
Some important baselines, such as adaptor-based methods and advanced neural operators, are missing from the experiments.

Recommendation:

The paper presents a promising approach for improving the generalization of neural PDE solvers. While the technical novelty is low, the paper's strengths in terms of clarity, empirical results, and hybrid framework outweigh the weaknesses. All the reviewers vote for acceptance. I, therefore, recommend accepting the paper and encourage the authors to use the feedback provided to improve the paper.